# Regularized Conditional Diffusion Model for Multi-Task Preference Alignment

**Xudong Yu**
Harbin Institute of Technology
hit20byu@gmail.com

**Chenjia Bai**[*]
Institute of Artificial Intelligence (TeleAI), China Telecom
baicj@chinatelecom.cn

**Haoran He**
Hong Kong University of Science and Technology
haoran.he@connect.ust.hk

**Changhong Wang**
Harbin Institute of Technology
cwang@hit.edu.cn

**Xuelong Li**
Institue of Artificial Intelligence (TeleAI), China Telecom
xuelong_li@ieee.org

## Abstract

Sequential decision-making can be formulated as a conditional generation process, with targets for alignment with human intents and versatility across various tasks. Previous return-conditioned diffusion models manifest comparable performance but rely on well-defined reward functions, which requires amounts of human efforts and faces challenges in multi-task settings. Preferences serve as an alternative but recent work rarely considers preference learning given multiple tasks. To facilitate the alignment and versatility in multi-task preference learning, we adopt multi-task preferences as a unified framework. In this work, we propose to learn preference representations aligned with preference labels, which are then used as conditions to guide the conditional generation process of diffusion models. The traditional classifier-free guidance paradigm suffers from the inconsistency between the conditions and generated trajectories. We thus introduce an auxiliary regularization objective to maximize the mutual information between conditions and corresponding generated trajectories, improving their alignment with preferences. Experiments in D4RL and Meta-World demonstrate the effectiveness and favorable performance of our method in single- and multi-task scenarios.

## 1 Introduction

In sequential decision-making, agents are trained to accomplish fixed or varied human-designed goals by interacting with the environment or learning from offline data. Two key objectives emerge during training: *alignment*, wherein agents are expected to take actions conforming to human intents expressed as crafted rewards or preferences, and *versatility*, denoting the capacity to tackle multiple tasks and generalize to unseen tasks. A promising avenue involves framing sequential decision-making as a sequence modeling problem via transformer [1] or diffusion models [2, 3]. This paradigm uses expressive models to capture the trajectory distributions and prevents unstable value estimation in conventional offline Reinforcement Learning (RL) methods [4]. Particularly, utilizing return-conditioned or value-guided diffusion models to perform planning or trajectory generation achieves favorable performance in offline RL [5, 6] and the multi-task variant [7, 8].

---

[*]Correspondence to: Chenjia Bai (baicj@chinatelecom.cn).

38th Conference on Neural Information Processing Systems (NeurIPS 2024).

Despite the great progress, applying diffusion models to sequential decision-making still faces challenges. (i) The condition generation process relies on a pre-defined reward function to provide return conditions. However, developing multiple task-specific reward functions in multi-task settings requires significant efforts [9] and may cause unintended behaviors [10]. (ii) The condition generation process with classifier-free guidance [11] often fails to ensure consistency between conditions and generations. As an example in Figure 1, the return-conditioned Decision Diffuser [5] struggles to achieve effective alignment for generated trajectories. Inspired by recent works [9, 12, 13], we find that preferences offer more versatile supervision across multi-tasks than scalar rewards. Specifically, the trajectories from the $i$-th task are preferred over the $j$-th task if we set the $i$-th task as the

target task. Conversely, we can reverse the preference label when setting the target task to the $j$-th task. Therefore, we adopt preferences rather than rewards to guide the conditional diffusion models in offline multi-task RL. The generated trajectories are intended to align with preferences, prioritizing higher returns or specific tasks.

To establish aligned and versatile conditional generation, our proposition involves adopting multi-task preferences and constructing a unified preference representation for both single- and multi-task scenarios, instead of learning scalar rewards from preference data [15, 16]. The acquired representations are aligned with the preference labels. The representations subsequently serve as conditions to guide the conditional generation of diffusion models. In this case, two key challenges arise: (i) *aligning the representations with preferences*, and (ii) *aligning the generated trajectories with representation conditions.*

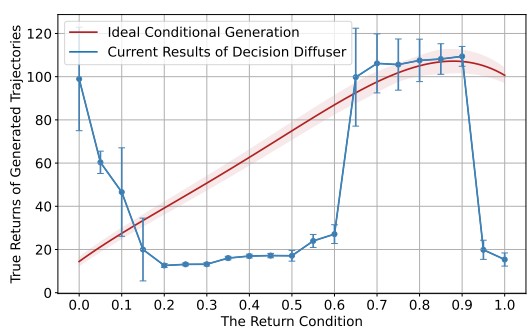

Figure 1: Illustration of the return-conditioned generation of Decision Diffuser [5] in *hopper-medium-expert* task. Existing return-conditional diffusion models fail to align generated trajectories with the return condition, while the red line indicates the desired relationship between the return conditions and true returns of generated trajectories [14].

To address the above challenges, we introduce Conditional Alignment via Multi-task Preference representations (CAMP) for multi-task preference learning. Specifically, we define multi-task preferences and extract preference representations from trajectory segments. (i) To align the representation with preferences, we propose a triplet loss and a Kullback-Leibler (KL) divergence loss to enable preferred and undesirable trajectories mapped into distinct zones in the representation space. The learned representations not only differentiate trajectories yielding higher returns but also discern trajectories across various tasks. Additionally, we learn an 'optimal' representation for each task to represent the trajectory with the highest preference. (ii) To align the generated trajectories with representation conditions in diffusion models, we introduce a Mutual Information (MI) regularization method. It augments representation-conditioned diffusion models with an auxiliary MI optimization objective, maximizing the correlation between the conditions and the generated outputs. During the inference for a specific task, we provide the 'optimal' representation for that task as the condition of the diffusion model, allowing the generated trajectories to adhere to desired preferences. Experiments on D4RL [17] and Meta-World [18] demonstrate the superiority of our method and the aligned performance in both multi-task and single-task scenarios.

## 2 Preliminaries

### 2.1 Diffusion Models for MDP

Diffusion models have emerged as effective generative models adept at learning high-dimensional data distribution $p(x)$ [2, 19, 20]. A diffusion model comprises a pre-defined forward process $q(x_k|x_{k-1}) = \mathcal{N}(x_k; \sqrt{\alpha_k}x_{k-1}, (1 - \alpha_k)I)$ and a trainable reverse process $p_\theta(x_{k-1}|x_k)$, where $k \in [1, K]$ denotes the timestep index, and $\alpha_k \in \mathbb{R}$ decides the variance schedule. By sampling Gaussian noise from $p(x_K)$ and iteratively employing the denoising step $p_\theta(x_{k-1}|x_k)$ for $K$ steps, diffusion models can generate $x_0 \sim p(x)$. Moreover, if additional conditions $c$ are introduced into

the denoising process such that $x_{k-1} \sim p_\theta(x_{k-1}|x_k, c)$, diffusion models can also estimate the conditional distribution $p(x|c)$.

The Markov Decision Process (MDP) is defined by a tuple $(\mathcal{S}, \mathcal{A}, P, r, \gamma)$, where $\mathcal{S}$ and $\mathcal{A}$ are the state and action spaces, $P$ is the transition function, $r$ is the reward function, and $\gamma$ is a discount factor. We consider an offline setting where the policy is learned from a given dataset $\mathcal{D}$. For multi-task learning, we introduce an additional task space $\mathcal{T}$ that contains $m$ discrete tasks. Diffusion models have been used for offline RL to overcome the distribution shift caused by Temporal-Difference (TD) learning [4]. Specifically, a diffusion model can formulate sequential decision-making as a conditional generative modeling problem by considering $x_0$ as a state sequence $(s_t, \ldots, s_{t+h})$ for planning [5]. The condition $c$ typically encompasses the return along the trajectory and is designed to generate trajectories with higher returns.

Optimizing the conditional diffusion model involves maximizing the log likelihood $\log p(x) = \log \int p(x|c)p(c)dc$, where $p(x|c)$ is the conditional likelihood for a specific $c$. Building on prior research [19, 21], the optimization is achieved by maximizing the Evidence Lower Bound (ELBO):

$$
\begin{aligned}
\log p(x_0) \geq \mathcal{L}_{\text{elbo}}(x_0, c) &\triangleq \mathbb{E}_{q(x_1|x_0)}[\mathbb{E}_{q_\psi}[\log p_\theta(x_0|x_1, c)]] - D_{\text{KL}}(q_\psi \| p(c)) \\
&- D_{\text{KL}}(q(x_K|x_0) \| p(x_K)) - \sum_{k=2}^{K} \mathbb{E}_{q(x_k|x_0)}[\mathbb{E}_{q_\psi}[D_{\text{KL}}(q_{x_{k-1}} \| p_\theta(x_{k-1}|x_k, c))]],
\end{aligned}
\tag{1}
$$

where $q_\psi := q_\psi(c|x_0)$ represents the approximate variational posterior mapping $x_0$ to the condition $c$, and $q_{x_{k-1}} = q(x_{k-1}|x_k, x_0)$. The complete derivation is provided in §A.1. In practice, this optimization problem can be addressed via a score-matching objective [2, 11] as,

$$
\mathcal{L}_\theta = \mathbb{E}[\|\epsilon - \epsilon_\theta(x_k, (1-\beta)c + \beta\varnothing, k)\|^2],
\tag{2}
$$

where $\epsilon_\theta$ is parameterized by a neural network to predict the noise $\epsilon \sim \mathcal{N}(0, I)$, the expectation is calculated w.r.t. $k \in [1, K], \tau \in \mathcal{D}$, and $\beta \sim \text{Bernoulli}(p)$.

## 2.2 Preference Learning for Conditional Generation

In decision-making, human preferences are often applied on trajectory segments $\tau = [s_i, a_i]_{i \in [1, h]}$ [22]. For a trajectory pair $(\tau_1, \tau_2)$, human feedback yields a preference label $y \in \{0, 1, 0.5\}$ that indicates which segment a human prefers. Here, $y = 1$ signifies $\tau_1 \succ \tau_2$, $y = 0$ signifies $\tau_1 \prec \tau_2$, and $y = 0.5$ means that two trajectories have the same preference. Previous studies commonly employ the Bradley-Terry (BT) model [23] to derive a reward function $\hat{r}$ from such preferences. Considering learning with offline preference data [16], and given a dataset $\mathcal{D}_\tau = \{(\tau_1, \tau_2, y)\}$, $\hat{r}$ is learned by maximizing the following objective:

$$
\mathcal{L}_{\hat{r}} = \mathbb{E}_{\mathcal{D}_\tau} \left[ y \log P[\tau_1 \succ \tau_2] + (1-y) \log P[\tau_2 \succ \tau_1] \right].
$$

In what follows, we simplify the notation by omitting the label $y$ and denote the preference data as $\{(\tau^+, \tau^-)\}$, where we have $\tau^+ \succ \tau^-$. Previous methods [24, 15] based on the BT model follow a two-phase learning process, first deriving a reward function from preference data and then training a policy with RL. Nevertheless, this process hinges on the assumption that pairwise preferences can be substituted with a reward model that can generalize to out-of-distribution data [25]. Moreover, these methods often require complex reward models [15] when preferences are intricate [26, 27].

As a result, we bypass the reward learning process and learn a preference-relevant representation that aligns well with trajectories in both single- and multi-task settings. Then the diffusion models can use these representations as conditions to generate trajectories that align with human preferences. In this setup, we regard $x_0 = \tau$ as the trajectory segments, and the condition $c$ as the preference-related representation of trajectories, denoted as $w = f(\tau)$. Therefore, the learning objective of diffusion model becomes $\log p(\tau_0)$, and the loss function becomes $\mathcal{L}_\theta = \mathbb{E}[\|\epsilon - \epsilon_\theta(\tau_k, (1-\beta)w + \beta\varnothing, k)\|^2]$.

## 3 Method

In this section, we give the definition of multi-task preferences and introduce how to extract preference representation from pairwise trajectories. Then, we present the conditional generation process and an auxiliary optimization objective to align the generated trajectories with preferences.

### 3.1 Versatile Representation for Multi-Task Preferences

**Multi-task preferences.** In the context of multi-task settings, tasks exhibit distinct reward functions, making reward-relevant information insufficient for providing versatile preferences across tasks. For example, moving fast is preferred and obtains high rewards in a 'running' task, while being unfavorable in a 'yoga' task. To address this challenge, we extend single-task preferences that only contain reward-relevant information to *multi-task* settings. Specifically, we consider two kinds of preferences given trajectories from $m$ tasks. For a specific task $i \in [m]$, trajectories are assessed based on (i) the **return** of trajectories when they belong to the same task, i.e., $\tau^{i+} \succeq \tau^{i-}$ if $\mathcal{R}(\tau^{i+}) \geq \mathcal{R}(\tau^{i-})$, where $\mathcal{R}(\cdot)$ calculates the cumulative reward, and (ii) the **task-relevance** of trajectories, i.e., $\tau^i \succ \tau^j$ with $j \neq i$. This means that trajectories from the target task $i$ are more preferred than any trajectories $\tau^j$ from a different task $j \in [m]$.

**Preference representations.** Based on the multi-task preferences, we propose to learn trajectory representations aligning with the preference data, as shown in Figure 2. The learned representations integrate both the trajectory and preference information, serving as the condition for subsequent trajectory generation. During learning representations, we also need to find the 'optimal' representations $\{w_i^*\}_{i \in [m]}$ that represent the optimal trajectories $\{\tau_i^*\}_{i \in [m]}$ for each task, where $\tau_i^*$ is preferred over any offline trajectories in task $i$. The $\{w_i^*\}_{i \in [m]}$ will be used for inference in diffusion models to generate desired trajectories for each task. Thus, we need to learn a trajectory encoder $w = f_\psi(\tau)$ that extracts preference-relevant information and the optimal representation $\{w_i^*\}_{i \in [m]}$.

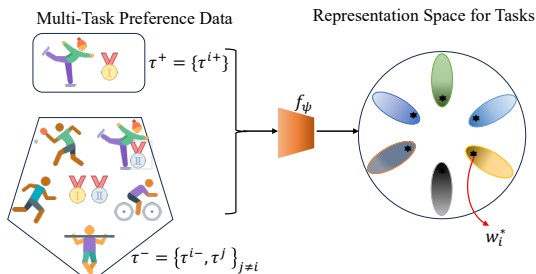

Figure 2: Illustration of the representation space of trajectories in multi-task preference data. For each task $i$, the positive samples $\tau^+$ consist of preferred trajectories $\tau^{i+}$ from task $i$, while negative samples $\tau^-$ include less preferred $\tau^{i-}$ from the same task, as well as $\tau^j$ from other tasks. Trajectories from diverse tasks are expected to be differentiated in the representation space, and $\{w_i^*\}_{i \in [m]}$ attempts to characterize the best trajectories for each task.

Furthermore, we model the representations from a distributional perspective to cover their uncertainty. In practice, the distribution of the optimal representation $p(w_i^*) = \mathcal{N}(\mu_i^*, \Sigma_i^*)$ and the distribution of representations given a trajectory $p(w|\tau) = \mathcal{N}(\mu_\psi(\tau), \Sigma_\psi(\tau))$ are both modeled as multivariate Gaussian distributions with a diagonal covariance matrix, where $\psi$ is parameterized by a transformer-based encoder. To summarize, the learnable parameters include vectors $\{\mu_i^*, \Sigma_i^*\}_{i \in [m]}$ for $\{w_i^*\}_{i \in [m]}$ and a trajectory encoder $f_\psi$.

**Loss functions for $f_\psi$ and $w_i^*$.** For each task $i$ in training, we build the multi-task dataset $\mathcal{D} = \{\tau^{i+}, \tau^{i-}, \tau^j\}$ to learn the representation space. The preference data are constructed by using the intra-task preference (i.e., $\tau^{i+} \succeq \tau^{i-}$) and the inter-task preference (i.e., $\tau^{i+} \succ \tau^j$). We denote the representation distributions as

$$p(w_i^+) = f_\psi(\tau^{i+}), \quad p(w_i^-) = f_\psi(\tau^{i-}) \text{ or } f_\psi(\tau^j).$$

For simplicity, we denote $p(w_i^+) = \mathcal{N}(\mu_i^+, \Sigma_i^+)$ and $p(w_i^-) = \mathcal{N}(\mu_i^-, \Sigma_i^-)$ by omitting the parameter $\psi$. These distributions are optimized using the following KL loss,

$$\mathcal{L}(\psi, \mu_i^*, \Sigma_i^*) = D_{\text{KL}}(\mathcal{N}(\mu_i^+, \Sigma_i^+)\|\mathcal{N}(\mu_i^*, \Sigma_i^*)) + 1/D_{\text{KL}}(\mathcal{N}(\mu_i^-, \Sigma_i^-)\|\mathcal{N}(\mu_i^*, \Sigma_i^*)). \quad (3)$$

This loss function encourages the encoder to map trajectories with similar preferences to closer embeddings while distancing dissimilar trajectories. In practice, we find that optimizing two sets of parameters (i.e., $\{\mu_i^*, \Sigma_i^*\}$ and $\psi$) simultaneously is unstable and leads to a trivial solution. Thus, we adopt an iterative optimizing process by using a stop-gradient (SG) operator. Specifically, we use the loss $\mathcal{L}(\text{SG}(\psi), \mu_i^*, \Sigma_i^*)$ to optimize $\{\mu_i^*, \Sigma_i^*\}$, and $\mathcal{L}(\psi, \text{SG}(\mu_i^*, \Sigma_i^*))$ to optimize the encoder $f_\psi$.

Furthermore, simply minimizing the KL loss cannot prevent the divergence of the unbounded term $D_{\text{KL}}(\mathcal{N}(\mu_i^-, \Sigma_i^-)\|\mathcal{N}(\mu_i^*, \Sigma_i^*))$, which may result in deviated representation distributions and unstable training. Hence, we add a triplet loss to learn representations, as

$$\mathcal{L}(\psi, \mu_i^*) = \mathbb{E}_{\mathcal{D}}[\max(d(\mu_i^+, \mu_i^*) - d(\mu_i^-, \mu_i^*) + \delta, 0)], \quad (4)$$

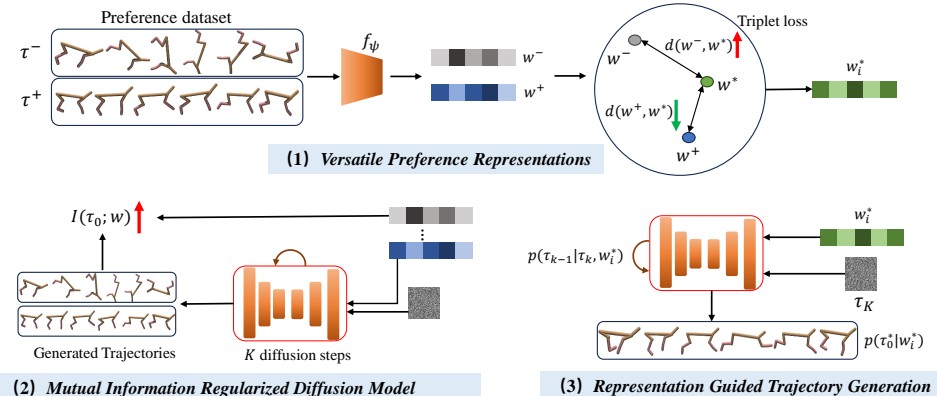

**Figure 3:** Overview of our method. (1) We learn preference representations $w = f_\psi(\tau)$ and the optimal one $w_i^*$ from trajectory segments $\tau$, which comprise positive samples $\tau^+$ and negative samples $\tau^-$. (2) We augment the diffusion model with an auxiliary mutual information term $I(\tau_0; w)$ to ensure the alignment between $\tau_0$ and $w$. (3) During the inference process, the diffusion model conditioned on $w_i^*$ can generate desired trajectories aligned with preferences.

where $d$ is the Euclidean distance in the embedding space. The triplet loss calculates the similarity between $w_i^*$ and preferred representations $w_i^+$, and the similarity between $w_i^*$ and unfavorable ones $w_i^-$, respectively. By regulating the distances between $\{\mu_i^+, \mu_i^-\}$ and $\mu_i^*$, the training process is stabilized. Meanwhile, minimizing the triplet loss also guarantees that the optimal embedding is more similar to $w_i^+$ and less similar to $w_i^-$, while $w_i^+$ and $w_i^-$ stay away from each other. We set $\delta$ as a margin between the two similarities and adopt the same iterative optimizing process for $\mathcal{L}(\psi, \mu_i^*)$. The illustration in Figure 3 captures our learning process. For brevity, we have omitted details related to the distributed form and multi-task learning components while retaining the core methodology.

**Prompting with $w_i^*$.** By optimizing the triplet and KL loss, $w_i^*$ gradually aligns with more preferred representations, converging to the optimal representations. After training, the diffusion model can be prompted by $(\mu_i^*, \Sigma_i^*)$ for a specific task $i$. The model will then generate the optimal trajectory for task $i$ with the guidance of conditions. The task $i$ can be a novel task not present in the training set, and the diffusion model will try to generalize to the new task in the representation space.

## 3.2 Representation Aligned Trajectory Generation

Given preference-based versatile representations, we train a diffusion model to generate trajectories that align with the representations. Prior work utilizes classifier-free guidance and aligns the generated samples with low-dimensional conditions, such as returns. However, we find this is insufficient to capture the complex relationship between conditions and trajectories. Using the representative method Decision Diffuser [5] as an example, Figure 1 reveals the relationship between the return conditions and true returns of generated trajectories in the *Hopper-medium-expert* task. While it is desirable to generate trajectories with higher returns as we increase the return condition, in practice, the attributes of generated samples do not exhibit a smooth transition with changes in the condition. Empirically, Decision Diffuser uses a target return of 0.8-0.9 for the generation process. This difficulty becomes more severe in our method because we cannot exclusively search the high-dimensional representation space and find an empirically effective representation. Therefore, it is critical to enhance the alignment between generated trajectories and previously learned preference representations.

**MI regularization.** Inspired by recent works on generative models [28, 29, 21], we introduce an auxiliary optimization objective aimed at strengthening the association between representations $w \sim q_\psi(w|\tau)$ and the generated trajectories $\tau_0$. Specifically, we augment the learning objective from Eq. (2) with a regularization term based on mutual information between $\tau_0$ and $w$. Formally, we train the conditional diffusion model using the following combined objectives:

$$\max \mathbb{E}_{q(\tau_0)}[\log p(\tau_0)] + \zeta \cdot I(\tau_0, w), \tag{5}$$

---

**Algorithm 1** Algorithm of CAMP

---

**Require:** Multi-task preferences $\mathcal{D}$, trajectory encoder $f_\psi(\cdot)$, diffusion model $\epsilon_\theta(\cdot)$, predictor $q_\phi(\cdot)$, inverse dynamic model $g_\omega(\cdot, \cdot)$, optimal representations $w_i^*$.
1: *// Training*
2: **for** each batch $\{\tau^{i+}, \tau^{i-}, \tau_j\}$ from $\mathcal{D}$ **do**
3:     Update encoder $f_\psi$ with $\mathcal{L}(\psi, \text{SG}(w_i^*))$ (ref. Eq. (4) & (3))
4:     Update optimal $w_i^*$ with $\mathcal{L}(\text{SG}(\psi), w_i^*)$ (ref. Eq. (4) & (3))
5:     Train MI-regularized diffusion model with Eq. (8).
6:     Update inverse dynamics model $g_\omega$ with Eq. (9).
7: **end for**
8: *// Inference*
9: **for** each step $t$ in one episode **do**
10:     Given current state $s_t$ and learned representations $w_i^*$.
11:     Generate trajectories $\tau = \{s_t, s_{t+1}, \cdots, s_{t+h}\}$ after $K$ denoising steps $p_\theta(\tau_{k-1}|\tau_k, w_i^*)$.
12:     Predict actions $\{a_t, \cdots, a_{t+h}\}$, where $a_t = g_\omega(s_t, s_{t+1})$.
13: **end for**

---

where $q(\tau_0)$ indicates the trajectory distribution of the offline dataset $\mathcal{D}$, and $\zeta$ is a hyper-parameter controlling the strength of regularization. Eq. (5) encompasses processes such as sampling trajectories $\tau \sim q(\tau_0)$, obtaining corresponding representations $w \sim q_\psi(w|\tau)$ via $f_\psi$, and the denoising process $p_\theta(\tau_{k-1}|\tau_k, w))$ to derive $\tau_0$.

**Tractable objective.** In practice, sampling $\tau_0$ from $p(\tau_0)$ to maximize the MI objective requires the diffusion model to denoise $K$ steps from $\tau_K$. This process imposes huge computational burden and may lead to potential memory overflow due to the gradient propagation across multi-steps. Consequently, there arises a need for an approximate objective, and an alternative approach is to replace the optimization on $I(\tau_0; w)$ with $I(\tau_k; w)$. This substitution is motivated by considering the relationship between $\tau_0$ and $\tau_k$, as described by the diffusion process $\tau_k = \sqrt{\bar{\alpha}_t}\tau_0 + \sqrt{1 - \bar{\alpha}_t}\epsilon_0 :=$ $f_{\text{diffuse}}(\tau_0)$. According to the data processing inequality [30], we have

$$I(\tau_0; w) \geq I(f_{\text{diffuse}}(\tau_0); w) = I(\tau_k; w), \tag{6}$$

As a result, $I(\tau_k; w)$ serves as a lower bound of $I(\tau_0; w)$, thus maximizing $I(\tau_k; w)$ also maximizes $I(\tau_0; w)$. In this case, Eq. (5) can be rewritten as:

$$\max \mathbb{E}_{q(\tau_0)}[\log p(\tau_0)] + \zeta \cdot I(\tau_k, w). \tag{7}$$

We note that the objective in Eq. (7) can be rewritten into an equivalent form that can be optimized efficiently (see §A.2 for a detailed derivation).

**Proposition 3.1.** *The optimization objective in Equation* (7) *can be transformed to*

$$\mathcal{L}_{\text{I}}(\theta, \phi) = \mathbb{E}_{q(\tau_0)}[\mathcal{L}_{\text{elbo}}(\tau_0, w)] - \zeta \cdot \mathbb{E}_{p(\tau_k)}\left[D_{KL}(p_\psi(w)\|q_\phi(w|\tau_k))\right]. \tag{8}$$

The first term resembles the ELBO term in Eq. (1) denoted as $\mathcal{L}_{\text{elbo}}(x_0 = \tau_0, c = w)$, which is the same as standard conditional diffusion models in Decision Diffuser [5]. The ELBO term aims to estimate the trajectory distribution via a conditional diffusion process, thus we adopt a similar conditional score-matching objective to optimize it. The second term contributes to enhancing the alignment between $\tau_0$ and $w$, where $p_\psi(w)$ is empirically averaged on samples $\tau \in \mathcal{D}$ via $f_\psi$, and $p(\tau_k) \propto f_{\text{diffuse}}(\tau_0)$ with $\tau_0$ sampling from $q(\tau_0)$. We can minimize the KL divergence to optimize $q_\phi(w|\tau_k)$, a variational predictor to predict the condition $w$ from the denoised sample $\tau_k$. In practice, we instantiate it with a neural network represented by $\phi$, taking predicted noises $\epsilon_\theta(\tau_k)$ as inputs.

### 3.3 Algorithmic Description

The entire procedure is shown in Algorithm 1. During training, we iteratively update the representation encoder $f_\psi$ and the optimal representation $w_i^*$ to learn versatile preference representations. Then we update the parameters of the conditional diffusion model via the loss function in Eq. (8). To decode the actions from a predicted trajectory, an inverse dynamic model is learned by using a supervised loss from the dataset, as

$$\min_\omega \mathbb{E}_{s,a,s'\sim\mathcal{D}} \|a - g_\omega(s, s')\|_2^2. \tag{9}$$

For planning in a specific task $i$, we use the optimal representation $w_i^*$ and the current state $s_t$ as a condition to generate an optimal trajectory $[s_t, \ldots, s_{t+h}]$. Then the inverse dynamics model $a_t = g_\omega(s_t, s_{t+1})$ is used to decode the action from two consecutive states.

## 4 Related Work

**Diffusion Models in RL.** Diffusion models exhibit significant advantages in generative modeling and characterizing complex distributions. On the one hand, they can be applied for modeling policies, with many research works [31–36] suggesting that diffusion models are more effective at capturing multi-modal action distributions than other policy types like energy-based policies. On the other hand, diffusion models are adept at directly predicting trajectories via conditional generation, adopting conditions such as value gradients [3, 37, 7], returns [5], or prompts [8]. Our work also trains diffusion models to generate trajectories, but focuses on guidance from preference representations. Regarding alignment with human preferences, recent works include the integration with designed attributes for various behaviors [6], fine-tuned policy with preferences [38, 39], and preference augmentation techniques to improve trajectory generation [40]. While these works focus on single-task settings, our work seeks a versatile solution by considering multi-task preferences.

**Learning From Human Preferences.** Current methods of learning from preferences can be categorized into two groups: *direct learning* and *indirect learning*. *Indirect learning* methods involve learning a reward model and incorporating existing RL algorithms. They employ techniques like off-policy learning [24], pseudo-labeling and data augmentation [41], iterative updates of reward models and policies [42], or leveraging Transformer architectures to learn reward models [15]. On the other hand, *direct learning* methods bypass reward model learning and directly incorporate human preferences into the learning process. This approach involves various strategies, such as combining decision transformer styles with preference embeddings [43], mapping reward functions to optimal policies [44, 45], or aligning models using extended Bradley-Terry comparisons [46]. This work falls under the category of offline *direct learning* approaches, framing it as a sequence modeling problem.

Our work considers challenging multi-task settings. While much prior research attempts to find Pareto optimal solutions considering the trade-offs between different preferences, these methods often require heuristic vector selection for unknown Pareto fronts [47, 48], millions of pre-collected preference labels and further online queries [10], or combination with Gaussian processes to learn preference relations [49]. In contrast, our approach does not require any heuristic methods and learns trajectory representations aligned with preferences from offline data.

## 5 Experiments

In this section, we will validate our approaches through extensive experiments. Our focus revolves around addressing the following key questions: (**i**) Can our approach demonstrate superior performance compared to existing approaches? (**ii**) Does the trajectory encoder discern different trajectories corresponding to multi-task preferences? And does $\{w_i^*\}_{i \in [m]}$ align with the optimal trajectories? (**iii**) Can the diffusion model capture the trajectory distribution and generate trajectories aligned with preferences? (**iv**) How about the generalization ability of our method on unseen tasks? (**v**) To what extent are multi-dimensional representations and regularization crucial to our approach?

### 5.1 Setups and Baselines

We conduct experiments on two benchmark datasets, **Meta-World** [18] for multi-task scenarios and **D4RL** [17] for single-task scenarios. Within evaluations on MetaWorld, we consider two distinct datasets: *near-optimal* dataset, which comprises the entire experiences obtained during the training of a SAC [50] agent for each task, and *sub-optimal* dataset, encompassing only the initial $50\%$ of the replay buffer. More details about datasets and baselines are provided in §B.

Categorically, our baselines encompass three types of approaches: **offline preference-based methods**, **offline reward-based RL methods**, and **behavior cloning** (BC). Within the realm of offline preference-based methods, our selections include: 1) **PT** [15], using a transformer network to model reward functions, integrated with RL algorithms; 2) **OPRL** [16], which employs ensemble-based reward functions; 3) **OPPO** [43], adopting a one-step process to model offline trajectories and

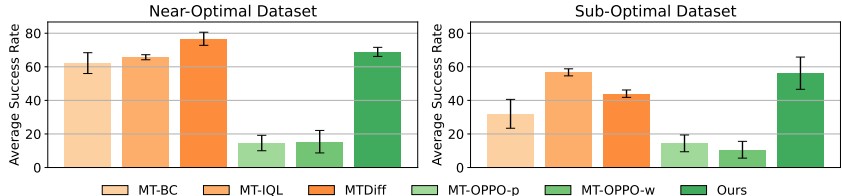

Figure 4: Average success rates in MT-10 benchmarks trained with different datasets. Orange bars are reward-based methods, while green bars represent preference-based methods. Detailed comparisons for each task can be found in §D.

Table 1: Performance comparison in D4RL benchmarks with scripted preferences. The subscript $\diamondsuit$ indicates the baseline with access to true reward functions, while $\spadesuit$ and $\clubsuit$ indicate the reported scores and our re-implementation with default parameters, respectively.

| Environments | BC | IQL$_\diamondsuit$ | PT | OPRL | OPPO$_\spadesuit$ | OPPO$_\clubsuit$ | CAMP (Ours) |
|---|---|---|---|---|---|---|---|
| halfcheetah-medium | $42.4 \pm 0.2$ | $48.3 \pm 0.2$ | $47.6 \pm 0.1$ | $42.0 \pm 2.8$ | $43.4 \pm 0.2$ | $42.5 \pm 0.2$ | $45.0 \pm 0.3$ |
| halfcheetah-medium-replay | $35.7 \pm 2.3$ | $44.5 \pm 0.2$ | $42.3 \pm 2.0$ | $41.5 \pm 2.6$ | $39.8 \pm 0.2$ | $33.6 \pm 2.4$ | $40.5 \pm 2.0$ |
| halfcheetah-medium-expert | $56.0 \pm 7.4$ | $94.7 \pm 0.5$ | $86.8 \pm 4.6$ | $90.5 \pm 4.0$ | $88.9 \pm 2.3$ | $89.6 \pm 0.9$ | $95.0 \pm 2.1$ |
| walker2d-medium | $63.3 \pm 16.2$ | $80.9 \pm 3.2$ | $76.8 \pm 6.5$ | $60.3 \pm 11.1$ | $85.0 \pm 2.9$ | $71.5 \pm 5.8$ | $73.9 \pm 0.8$ |
| walker2d-medium-replay | $21.8 \pm 10.2$ | $82.2 \pm 3.0$ | $75.7 \pm 3.9$ | $53.3 \pm 6.2$ | $71.7 \pm 4.4$ | $19.7 \pm 10.3$ | $60.5 \pm 1.1$ |
| walker2d-medium-expert | $99.0 \pm 16.0$ | $111.7 \pm 0.9$ | $110.4 \pm 0.5$ | $105.4 \pm 5.6$ | $105.0 \pm 2.4$ | $97.8 \pm 19.1$ | $104.8 \pm 3.0$ |
| hopper-medium | $53.5 \pm 1.8$ | $67.5 \pm 3.8$ | $25.7 \pm 1.2$ | $45.6 \pm 3.5$ | $86.3 \pm 3.2$ | $48.7 \pm 4.2$ | $59.3 \pm 0.8$ |
| hopper-medium-replay | $29.8 \pm 2.1$ | $97.4 \pm 6.4$ | $82.0 \pm 7.9$ | $45.6 \pm 10.9$ | $88.9 \pm 2.3$ | $29.7 \pm 17.1$ | $56.2 \pm 0.7$ |
| hopper-medium-expert | $52.3 \pm 4.0$ | $107.4 \pm 7.8$ | $44.0 \pm 8.8$ | $68.8 \pm 18.1$ | $108.0 \pm 5.1$ | $99.5 \pm 23.5$ | $108.9 \pm 1.0$ |
| Average | 50.4 | 81.6 | 65.7 | 61.4 | 79.7 | 59.2 | 71.6 |

preferences, avoiding reward learning. For methods enjoying access to true reward functions, our selection includes: 1) **IQL** [4], which performs in-distribution Q-learning and achieves significant performance; 2) **MTDiff** [8], a method leveraging diffusion models in multi-task scenarios. Since many baselines do not consider multiple tasks, we make modifications and mark them with '**MT−**'.

## 5.2 Performance Comparison

**Multi-task Performance on Meta-World** We assess the multi-task performance using MT-10 tasks and present the results in Figure 4. Our observations are as follows: **1)** Given near-optimal datasets, CAMP outperforms MT-BC and MT-IQL trained with ground-truth rewards, with only a small performance gap compared to MTDiff. **2)** Given sub-optimal datasets, CAMP demonstrates comparable performance to MT-IQL and surpasses other baselines, highlighting its robustness to dataset quality. **3)** Two variations of OPPO fail to learn effective policies for multiple tasks, exposing the limitations of existing preference-based RL methods in handling multi-task scenarios. In comparison, CAMP achieves a performance improvement of nearly four times.

**Single-task Performance on D4RL** As shown in Table 1, CAMP demonstrates superior performance across all D4RL Mujoco tasks compared to BC. Additionally, it exhibits comparable performance with IQL in medium-expert tasks. When compared to offline preference-based methods such as PT or OPRL, CAMP showcases significant improvements, particularly in hopper tasks. While OPPO is an effective preference-based method, we observe its sensitivity to random seeds. In our re-implementation using default hyperparameters, its performance degrades in walker2d and hopper tasks. In contrast, CAMP provides aligned trajectory generation and favorable performance.

## 5.3 Visualization

While showing superior performance, does CAMP learn meaningful representations or perform desired conditional generation? In this part, we map trajectory segments to two-dimensional space via T-SNE [51] visualization and analyze properties of the trajectory encoder and the diffusion model.

**Do representations $w$ discern different trajectories?** To assess the capabilities of discerning different trajectories and aligning with the best trajectories, we sample several trajectories from $\mathcal{D}$ and project them to the latent space via $f_\psi$, with subsequent T-SNE visualization. The results at different stages during training are illustrated in the left panel of Figure 5. With increasing training steps, we find a gradually clear classification among trajectories with different returns. Meanwhile, the optimal representations $w_i^*$, represented as red dots, gradually approach trajectories with higher returns.

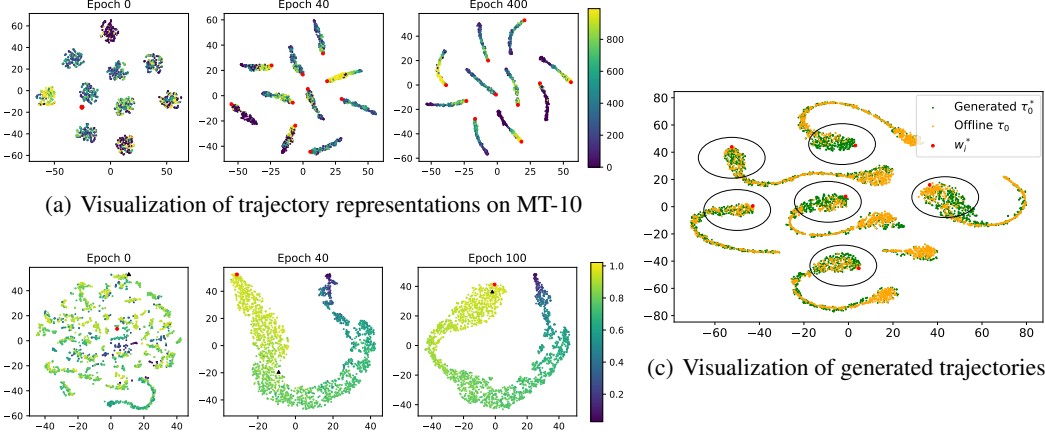

(a) Visualization of trajectory representations on MT-10

(c) Visualization of generated trajectories

(b) Visualization of representations on Hopper-medium-expert

Figure 5: **Left:** Brighter dots indicate trajectories with higher returns. Red dots represent each dimension of $w_i^*$. Black triangles in (b) mark trajectories with the highest return. $f_\psi$ can separate trajectories from different tasks and with different returns. $w_i^*$ aligns with the optimal trajectories for each task. **Right:** Guided by $w_i^*$, diffusion models can generate trajectories $\tau_0^*$ that mainly lie around $w_i^*$ (shown as black circles), which represents better trajectories in offline data $\tau_0$.

**Does the diffusion model generate trajectories aligned with preferences?** While $w_i^*$ provides useful guidance, we want to validate the generative ability of the conditional diffusion model. We map the generated trajectories to latent space and compare them with offline trajectories. As demonstrated in Figure 5(c), the green points denote the generated trajectories, while the red dots represent $w_i^*$. We observe that these generated trajectories for 6 tasks align closely with $w_i^*$, which represent the more favorable trajectories within the dataset. This phenomenon showcases that generated trajectories guided by $w_i^*$ align with preferred trajectories, validating our approach's effectiveness.

### 5.4 Analysis on Generalization Ability

To validate the generalization ability, we evaluate generated trajectories guided by $\hat{w}_k^* \notin \{w_i^*\}_{i \in [m]}$ that are learned from trajectories of unseen tasks. $\hat{w}_k^*$ from those new tasks are learned using the same method as described in Section 3.1. We hold the diffusion model trained on MT-10 tasks fixed and assess its performance when faced with unseen representation conditions. We compare its performance with that of MT-BC, MT-IQL, and MTDiff, all of which are trained using the same settings. As depicted in Table 2, our approach exhibits favorable performance on unseen tasks and outperforms baseline methods by a considerable margin. Further analyses are presented in §H.

Table 2: Generalization performance on five unseen tasks. CAMP exhibits superior performance.

| Unseen Tasks | MT-BC | MT-IQL | MTDiff | CAMP |
|---|---|---|---|---|
| button-press-wall-v2 | $0.0 \pm 0.0$ | $21.0 \pm 21.0$ | $16.0 \pm 17.3$ | $\mathbf{22.0 \pm 3.3}$ |
| button-press-topdown-wall-v2 | $21.0 \pm 19.0$ | $33.0 \pm 19.0$ | $72.0 \pm 8.6$ | $66.8 \pm 5.0$ |
| handle-press-v2 | $43.0 \pm 13.0$ | $60.0 \pm 4.0$ | $36.7 \pm 12.3$ | $\mathbf{76.8 \pm 7.2}$ |
| handle-press-side-v2 | $0.0 \pm 0.0$ | $1.0 \pm 1.0$ | $2.0 \pm 2.8$ | $\mathbf{11.6 \pm 1.5}$ |
| peg-unplug-side-v2 | $0.0 \pm 0.0$ | $0.0 \pm 0.0$ | $2.7 \pm 1.9$ | $1.6 \pm 1.5$ |
| Average | 12.4 | 19.9 | 17.8 | **35.8** |

### 5.5 Ablation Study

This part delves into the impact of multi-dimensional representations and the MI regularization term. Due to space limitations, detailed implementations refer to §C. Here, we highlight key conclusions: **1)** Learning a multi-dimensional representation and choosing a suitable dimension are crucial. When the dimension is too low, such as $|w| = 1$, the representations are insufficient to capture preferences in trajectories and provide effective guidance. Conversely, when the dimension is too high, as in $|w| = 64$, the representation space becomes challenging to learn, resulting in inferior performance. **2)** The auxiliary loss of mutual information plays a pivotal role in our framework. Without this regularization term, our method's performance on all MetaWorld and D4RL tasks degrades.

# 6   Conclusion

This paper introduces a regularized conditional diffusion model for alignment with preferences in multi-task scenarios. Based on the versatile multi-task preferences, our method acquires preference representations that differentiate trajectories across tasks and with different returns, as well as an optimal representation aligning with the best trajectory for each task. By regularizing exiting diffusion models in RL with mutual information maximization between conditions and generated trajectories, our method can generate desired trajectories by conditioning on the optimal representation for each task, ensuring alignment with preferences. Experimental validation demonstrates the favorable performance and generalization ability of our method. Future work may involve using faster sampling methods to enhance algorithm efficiency or extending to fine-tuning foundation models.

# Acknowledgments

This work is supported by the National Natural Science Foundation of China (Grant No.62306242) and the Yangfan Project of the Shanghai Municipal Science and Technology (Grant No.23YF11462200).

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

# A  Theoretical Analysis

## A.1  Derivation of ELBO in Equation (1)

Here we derive the ELBO of the diffusion model by considering a conditional denoising process $p_\theta(x_{k-1}|x_k, c)$:

$$
\begin{aligned}
\log p(x_0) &= \log \int p(c)p(x_{0:K}|c)dx_{1:K}dc \\
&= \log \int \frac{p(c)p(x_{0:K}|c)q(x_{1:K}|x_0)q_\phi(c|x_0)}{q(x_{1:K}|x_0)q_\phi(c|x_0)}dx_{1:K}dc \\
&= \log \mathbb{E}_{q(x_{1:K}|x_0)}\left[\mathbb{E}_{q_\psi(c|x_0)}\left[\frac{p(c)p(x_{0:K}|c)}{q(x_{1:K}|x_0))q_\psi(c|x_0))}\right]\right] \\
&\geq \mathbb{E}_{q(x_{1:K}|x_0)}\left[\mathbb{E}_{q_\psi(c|x_0)}\left[\log \frac{p(c)p(x_{0:K}|c)}{q(x_{1:K}|x_0))q_\psi(c|x_0))}\right]\right] \\
&= \mathbb{E}_{q(x_{1:K}|x_0)}\left[\mathbb{E}_{q_\psi(c|x_0)}\left[\log \frac{p(c)p(x_K)p_\theta(x_0|x_1,c)\prod_{k=2}^K p_\theta(x_{k-1}|x_k,c)}{q_\psi(c|x_0)q(x_1|x_0)\prod_{k=2}^K q(x_k|x_{k-1},x_0)}\right]\right] \\
&= \mathbb{E}_{q(x_{1:K}|x_0)}\left[\mathbb{E}_{q_\psi(c|x_0)}\left[\log \frac{p(c)}{q_\psi(c|x_0)}\cdot p_\theta(x_0|x_1,c)\cdot \frac{p(x_K)}{q(x_1|x_0)}\cdot \frac{\prod_{k=2}^K p_\theta(x_{k-1}|x_k,c)}{\prod_{k=2}^K q(x_k|x_{k-1},x_0)}\right]\right] \\
&= \mathbb{E}_{q(x_{1:K}|x_0)}\left[\mathbb{E}_{q_\psi(c|x_0)}\left[\log \frac{p(c)}{q_\psi(c|x_0)}\cdot p_\theta(x_0|x_1,c)\cdot \frac{p(x_K)}{q(x_1|x_0)}\cdot \frac{\prod_{k=2}^K p_\theta(x_{k-1}|x_k,c)q(x_{k-1}|x_0)}{\prod_{k=2}^K q(x_{k-1}|x_k)q(x_k|x_0)}\right]\right] \\
&= \mathbb{E}_{q(x_{1:K}|x_0)}\left[\mathbb{E}_{q_\psi(c|x_0)}\left[\log \frac{p(c)}{q_\psi(c|x_0)}\cdot p_\theta(x_0|x_1,c)\cdot \frac{p(x_K)}{q(x_K|x_0)}\cdot \frac{\prod_{k=2}^K p_\theta(x_{k-1}|x_k,c)}{\prod_{k=2}^K q(x_{k-1}|x_k,x_0)}\right]\right] \\
&= \mathbb{E}_{q(x_{1:K}|x_0)}\left[\mathbb{E}_{q_\psi(c|x_0)}\left[\log \frac{p(c)}{q_\psi(c|x_0)} + \log p_\theta(x_0|x_1,c) + \log \frac{p(x_K)}{q(x_K|x_0)} + \sum_{k=2}^K \log \frac{p_\theta(x_{k-1}|x_k,c)}{q(x_{k-1}|x_k,x_0)}\right]\right] \\
&= \mathbb{E}_{q_\psi(c|x_0)}\left[\log \frac{p(c)}{q_\psi(c|x_0)}\right] + \mathbb{E}_{q(x_1|x_0)}\left[\mathbb{E}_{q_\psi(c|x_0)}[\log p_\theta(x_0|x_1,c)]\right] + \mathbb{E}_{q(x_K|x_0)}\left[\log \frac{p(x_K)}{q(x_K|x_0)}\right] \\
&\quad + \sum_{k=2}^K \mathbb{E}_{q(x_{k-1},x_k|x_0)}\left[\mathbb{E}_{q_\psi(c|x_0)}\left[\log \frac{p_\theta(x_{k-1}|x_k,c)}{q(x_{k-1}|x_k,x_0)}\right]\right] \\
&= \underbrace{\mathbb{E}_{q(x_1|x_0)}\left[\mathbb{E}_{q_\psi(c|x_0)}[\log p_\theta(x_0|x_1,c)]\right]}_{\text{reconstruction term}} - \underbrace{D_{\mathrm{KL}}(q(x_K|x_0)||p(x_K))}_{\text{prior matching term for } x_K} - \underbrace{D_{\mathrm{KL}}(q_\psi(c|x_0)||p(c))}_{\text{prior matching term for } c} \\
&\quad - \sum_{k=2}^K \underbrace{\mathbb{E}_{q(x_k|x_0)}\left[\mathbb{E}_{q_\psi(c|x_0)}[D_{\mathrm{KL}}(q(x_{k-1}|x_k,x_0)||p_\theta(x_{k-1}|x_k,c))]\right]}_{\text{denoising matching term}}.
\end{aligned}
\tag{10}
$$

Following previous work [19, 21], the above ELBO provides interpretations for each term:

- The reconstruction term resembles the part of the ELBO of a vanilla variational autoencoder, and can be optimized using Monte Carlo estimates [19]. In [2], this term is learned using a separate decoder.

- The prior matching term for $x_K$ indicates the discrepancy between the distribution of the noisy version of $x_0$ after $K$ steps and the standard Gaussian prior. This term has no trainable parameters, and we ignore it during training.

- The denoising matching term measures the discrepancy between the ground-truth denoising function $q(x_{t-1}|x_t, x_0)$ and the approximated denoising transition function $p_\theta(x_{t-1}|x_t, c)$. It is minimized when the approximated denoising transition stays close to the ground-truth denoising transition step.

- The prior matching term for $c$ indicates the discrepancy between the approximate posterior $q_\psi(c|x_0)$ and the prior $p(c)$, which can be a standard Gaussian distribution.

## A.2 Derivation of the loss function in Equation (8)

First, we give several explanations about Eq. (7), which provides a lower bound of Eq. (5). This relationship is built on top of that $I(\tau_0; w) \geq I(\tau_k, w)$. Intuitively, $\tau_k$ consists of more Gaussian noises than $\tau_0$, thus providing less information about $w$. In particular, when $k = K$, we get a pure Gaussian noise so that $I(\tau_K; w) = 0$.

Then we analyze Eq. (7) in two parts and derive them to two terms in Eq. (8), respectively. For $\mathbb{E}_{q(\tau_0)}[\log p(\tau_0)]$, we rewrite Eq. (10) by setting $x_0 = \tau_0, c = w$ and average the terms over the data distribution $q(\tau_0)$:

$$
\begin{aligned}
\mathbb{E}_{q(\tau_0)}[\log p(\tau_0)] &\geq \mathbb{E}[\mathcal{L}_{\text{elbo}(\tau_0, w)}] \\
&= \mathbb{E}_{q(\tau_0, \tau_1)}\big[\mathbb{E}_{q_\psi(w|\tau_0)}\big[\log p_\theta(\tau_0|\tau_1, w)\big]\big] \\
&\quad - \mathbb{E}_{q(\tau_0)}\big[D_{\text{KL}}(q(\tau_K|\tau_0)\|p(\tau_K))\big] \\
&\quad - \sum_{k=2}^{K}\mathbb{E}_{q(\tau_{k-1}, \tau_k, \tau_0)}\big[\mathbb{E}_{q_\psi(w|\tau_0)}\big[D_{\text{KL}}(q(\tau_{k-1}|\tau_k, \tau_0)\|p_\theta(\tau_{k-1}|\tau_k, w))\big]\big] \\
&\quad - \mathbb{E}_{q(\tau_0)}[D_{\text{KL}}[q_\psi(w|\tau_0)\|p(w)]].
\end{aligned}
\tag{11}
$$

According to previous work [19], this optimization problem can be simplified as:

$$
\arg\min_\theta \frac{1}{2\sigma_q^2(k)} \frac{(1-\alpha_k)^2}{(1-\bar\alpha_k)\alpha_k}\left[\|\epsilon_0 - \epsilon_\theta(\tau_k, w, k)\|_2^2\right],
\tag{12}
$$

where $\sigma_q$ is a function of $\alpha$ coefficients and $\epsilon_0 \sim \mathcal{N}(\epsilon; 0, I)$ is the source noise that determines $\tau_k$ from $\tau_0$. For the mutual information regularization term $I(\tau_k, w)$, it can be derived as follows:

$$
\begin{aligned}
I(\tau_k; w) &= H(w) - H(w|\tau_k) \\
&= H(w) + \int\int p(w = w', \tau_k)\log p(w = w'|\tau_k)dw'd\tau_k \\
&= H(w) + \mathbb{E}_{p(\tau_k)}\left[\mathbb{E}_{p(w'|\tau_k)}\left[\log\frac{p(w'|\tau_k)}{q_\phi(w'|\tau_k)}q_\phi(w'|\tau_k)\right]\right] \\
&= H(w) + \mathbb{E}_{p(\tau_k)}\bigg[\underbrace{D_{\text{KL}}\left[p(w'|\tau_k)\|q_\phi(w'|\tau_k)\right]}_{D_{\text{KL}}\geq 0}\bigg] + \mathbb{E}_{p(\tau_k)}[\mathbb{E}_{p(w'|\tau_k)}[\log q_\phi(w'|\tau_k)]] \\
&\geq H(w) + \mathbb{E}_{p(\tau_k)}[\mathbb{E}_{p(w'|\tau_k)}[\log q_\phi(w'|\tau_k)]]
\end{aligned}
\tag{13}
$$

We introduce a lemma to derive the above inequality. Please refer to lemma 5.1 of [29] for detailed proofs.

**Lemma A.1.** *For random variables $X, Y$ and function $f(x, y)$ under suitable regularity conditions:* $\mathbb{E}_{x\sim X, y\sim Y|x}[f(x, y)] = \mathbb{E}_{x\sim X, y\sim Y, x'\sim X|y}[f(x', y)]$.

By using lemma A.1, we can derive that

$$
\begin{aligned}
I(\tau_k; w) &\geq H(w) + \mathbb{E}_{p(\tau_k)}[\mathbb{E}_{p(w'|\tau_k)}[\log q_\phi(w'|\tau_k)]] \\
&= H(w) + \mathbb{E}_{p_\psi(w)}\big[\mathbb{E}_{p(\tau_k|w)}\big[\mathbb{E}_{p(w'|\tau_k)}[\log q_\phi(w'|\tau_k)]\big]\big] \\
&= \mathbb{E}_{p_\psi(w)}\left[-\log p_\psi(w)\right] + \mathbb{E}_{p_\psi(w)}\big[\mathbb{E}_{p(\tau_k)}[\log q_\phi(w|\tau_k)]\big] \\
&= \mathbb{E}_{p(\tau_k)}\left[\mathbb{E}_{p_\psi(w)}\left[\log\frac{q_\phi(w|\tau_k)}{p_\psi(w)}\right]\right] \\
&= -\mathbb{E}_{p(\tau_k)}[D_{\text{KL}}[p_\psi(w)\|q_\phi(w|\tau_k)]]
\end{aligned}
\tag{14}
$$

In Eq. (14), we omit the condition of $p(\tau_k|w)$ as $p(\tau_k)$ because $\tau_k$ comes from $\tau_0 \sim \mathcal{D}$ by adding Gaussian noises. Combining Eqs. (11) and (14), we can obtain the tractable objective in Eq. (8).

# B  Implementation Details

In this part, we introduce some details about our methods and evaluation, including benchmarks, baselines, and implementations. We then provide specific comparisons with several related works, including Decision Diffuser, MTDiff, and OPPO.

## B.1  Datasets

**MetaWorld MT-10**  In this study, we employed MT-10 tasks from MetaWorld as benchmarks to assess multi-task performance. These tasks share similar dynamics, involving the manipulation of a Sawyer robot to interact with various objects to achieve diverse manipulation goals, as shown in Figure 6. Each task exhibits distinct state spaces and reward functions, presenting a significant challenge for learning strategies. The primary evaluation metric we utilized is the success rate of task completion, with the attained return serving as a secondary measure.

Considering this is focused on an offline learning setting, we followed the methodology of MTDiff [8], employing a replay buffer during SAC [50] training as the offline dataset. For each task, we trained an agent using SAC to progress from a random policy toward an expert policy. Additionally, we categorized two distinct datasets: the *near-optimal* dataset and the *sub-optimal* dataset, differing in the number of expert trajectories included. The near-optimal dataset comprises all the replay buffer data, totaling 100 million transitions, while the sub-optimal dataset contains only the initial 50% of this data.

Following previous work [43, 15, 16], We collect preference data with a scripted teacher. In particular, for multi-task preferences, we construct intra-task preferences among trajectories by comparing their returns and inter-task preferences among tasks based on the task relevance, as illustrated in Figure 2.

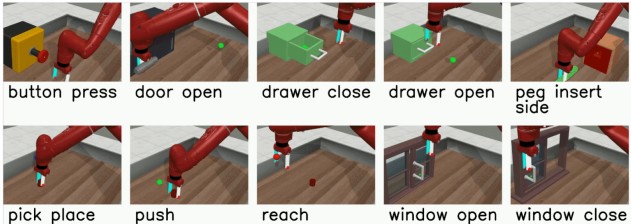

Figure 6: MetaWold MT-10 tasks. The goal is to learn a policy that can succeed on a diverse set of tasks.

**D4RL**  To evaluate our methods on single-task settings, we conduct experiments on the Mujoco locomotion tasks from D4RL benchmarks. Three tasks Halfcheetah, Walker2d, and Hopper are chosen, with three types of datasets, including medium, medium-replay, and medium-expert. We obtain preferences from scripted teachers, following previous work [15, 43, 16].

## B.2  Baselines

As outlined in Section 5.1, the baselines we compare can be categorized into three groups: offline preference-based methods, offline reward-based RL methods utilizing ground-truth reward functions, and behavior cloning. In the following sections, we delve into the specifics and experimental details of these baselines to provide a comprehensive comparative analysis.

**PT**  PT utilizes transformer architecture to learn a scalar reward model, which is then used to optimize policies with the IQL algorithm. We use its official implementation and default parameters [2].

**OPRL**  OPRL combines IQL with reward functions from ensemble-based disagreement. We use its official implementation and default parameters [3].

---

[2] https://github.com/csmile-1006/PreferenceTransformer
[3] https://github.com/danielshin1/oprl

**OPPO** & **MT-OPPO-p** & **MT-OPPO-w**     OPPO shares a similar motivation with us for avoiding explicitly reward learning. It models offline trajectories and preferences in a one-step process. Specifically, it optimizes an offline hindsight information matching objective for seeking a conditional policy and a preference modeling objective for finding the optimal condition. We use its official implementation and default parameters [4]. We observe unstable results over 5 random seeds, which is not the same as the reported results in the paper. We speculate that OPPO may be sensitive to different random seeds.

To make it suitable for multi-task settings, we make two types of modifications to its original version, named 'MT-OPPO-p' and 'MT-OPPO-w'. For 'MT-OPPO-p', we add task id inputs for the conditional policy, so that the original policy $\pi(a|s, z)$ can be extended to $\pi(a|s, z, \text{ID})$. For 'MT-OPPO-w', we modify OPPO using the same method as our approach, extending its representations to multiple dimensions, so that the policy becomes $\pi(a|s, \mathbf{z}, \mathbf{ID})$ and $\mathbf{z}$ is a multi-dimension representation. Following previous work [8], we project task IDs to latent variables via a 3-layer MLP.

**BC** & **MT-BC**     Traditional behavior cloning learns a direct mapping from states to actions $\pi(a|s)$. In our experiments, we encode the scalar task ID to a latent variable via MLP and concatenate the latent variable with the original states. Therefore, MTBC utilizes a conditional policy by conditioning on task IDs. We modify it based on implementations from CORL [5].

**IQL** & **MT-IQL**     IQL is an effective offline RL method, which performs in-distribution Q-learning and expectile regression. We use the implementation from CORL and make similar modifications to MTBC.

**MTDiff**     MTDiff is a diffusion-based method that combines Transformer backbones and prompt learning for generative planning in multi-task settings. We use its official implementation and default parameters [6].

### B.3     Our implementation

Our code is built on Decision Diffuser[7] and OPPO[8]. We leverage their implementations of diffusion models and transformer-based encoders while developing the representation learning process for multi-task scenarios and the auxiliary mutual information regularization.

- Following Decision Diffuser, we use the temporal U-Net architecture to predict noise, where timesteps and representations $w$ are separately mapped to 128-dimensional vectors by 2-layered MLPs.
- We employ 200 denoising steps, consistent with previous work [5, 8].
- The training details of the inverse dynamics model $g$ are aligned with those of Decision Diffuser.
- For the trajectory encoder, which projects trajectory segments $\tau$ to latent representations $w$, we adopt a similar Transformer architecture to that of OPPO, but we learn distributional representations.
- The conditional guidance weight in diffusion models is set to 1.2 for most tasks and 1.5 for the halfcheetah-medium-expert task.
- The learning rate of the diffusion model is $2e^{-4}$ with the Adam optimizer.
- Training steps are set to $2e^6$ in MetaWorld tasks and $1e^6$ in D4RL tasks, with results averaged over multiple seeds. In MetaWorld benchmarks, as each environment has 50 random goals, evaluations are averaged over these 50 random goals.
- The horizon $h$ of trajectories is set to 20 in the MT-10, halfcheetah, and walker2d tasks, and 100 in the hopper tasks.

---

[4]https://github.com/bkkgbkjb/OPPO
[5]https://github.com/tinkoff-ai/CORL
[6]https://github.com/tinnerhrhe/MTDiff
[7]https://github.com/anuragajay/decision-diffuser/
[8]https://github.com/bkkgbkjb/OPPO

- Batch size is set to 256 for halfcheetah and walker2d tasks, and 32 for hopper tasks and each task in MetaWorld MT-10 tasks (total batch size is 320).
- The regularization coefficient $\zeta$ is set to 0.1 for the MT-10, halfcheetah, and hopper-medium tasks, 0.5 for the walker2d-medium and walker2d-medium-expert tasks, 0.01 for the hopper-medium-replay and walker2d-medium-replay tasks, and 1.0 for the hopper-medium-expert task.
- The dimension of preference representations is set to 16.
- We conduct training on an NVIDIA GeForce RTX 3090. Training time varies with task complexity, approximately 30 hours for D4RL tasks and 59 hours for MT-10 tasks.

## B.4 Difference with other methods

**Decision Diffuser**   Decision Diffuser is an effective method for conditional generative modeling, with conditioning options including returns, constraints, or skills. In particular, when using returns as conditions, Decision Diffuser utilizes normalized returns and sets the target return as 0.9 for most tasks during the generation process. In Figure 1, we have revealed that the return-conditioned paradigm may not ensure alignment between specified return conditions and generated trajectories. Unlike Decision Diffuser, we propose a regularization term for conditional diffusion models and obtain enhanced alignment between given representation conditions and generated trajectories. As shown in Figure 5(c), generated trajectories $\tau_0^*$ under the guidance of $w_i^*$ cluster around the region near $w_i^*$. Our method also differs from Decision Diffuser in the utilization of preference data instead of reward labels and the versatility across multi-task scenarios.

**MTDiff**   MTDiff employs diffusion models for modeling large-scale multi-task offline data. Similar to Decision Diffuser, MTDiff adopts classifier-free guidance but introduces prompt learning for both modeling policies and trajectories. Specifically, it utilizes normalized cumulative returns and regards task-relevant information as prompts. By employing these task-specific prompts as conditions, MTDiff distinguishes between different tasks and generates desired trajectories for each specific task. The prompts consist of expert demonstrations in the form of trajectory segments, akin to the approach in PromptDT [52]. In contrast, our method eliminates the need for expert demonstrations and instead extracts multi-dimensional representations from multi-task preferences. These representations serve as conditions for the diffusion model. Experiments conducted on MetaWorld tasks demonstrate that our method achieves comparable performance to MTDiff, all without the necessity for reward labels or demonstration prompts.

**OPPO**   In the realm of offline preference-based reinforcement learning, OPPO models offline trajectories and preferences in a one-step process, circumventing the need for reward modeling. Specifically, OPPO learns an optimal context using a preference modeling objective and subsequently optimizes a contextual policy. During the learning of contexts, OPPO constructs positive and negative samples and uses the triplet loss to optimize the contexts. With these learned contexts, OPPO then develops a conditional policy, akin to DT [53] and RvS [54]. However, our method utilizes the KL loss and the triplet loss to optimize representation distributions, which align with multi-task preferences. Moreover, our approach focuses on conditional diffusion models and the alignment for trajectory generation. It is noteworthy that our method demonstrates more stable performance and excels in multi-task settings, exhibiting successful generalization to unseen tasks. In contrast, OPPO is specifically designed for the single-task setting.

## C   Ablation Study

In this part, we aim to dissect and analyze the influences of two key elements: 1) the dimension of representations, and 2) the auxiliary mutual information optimization objective.

### C.1   How does the dimension of $w$ affect our method?

When the dimension of $w$ is reduced to 1, the learned representations in our method resemble those of vectorized reward models [55] or distributional rewards [56]. Conversely, if the dimension is too high, both learning representations and aligning conditional generation become challenging

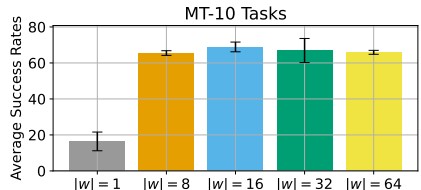

Figure 7: Ablation on the dimension of $w$.

due to the increased complexity of the representation space. We conduct an ablation study with $|w| \in 1, 8, 16, 32, 64$, where $|w| = 1$ and $|w| = 64$ represent two extremes. As presented in Figure 7, the performance on MT-10 tasks significantly decreases when $|w| = 1$, and it exhibits performance degradation when the dimension of representations is too big. This implies that we should adopt a suitable dimension for preference representations, and we choose $|w| = 16$ in our experiments.

## C.2 Is the mutual information regularization critical?

Our approach employs a regularization term based on mutual information to enhance the alignment between preference representations and generated trajectories. In this section, we investigate the impact of this regularization term on the final performance. Specifically, we set $\zeta = 0$ for evaluations on MT-10 and D4RL tasks. As illustrated in Table 3, discarding this mutual information regularization term results in performance degradation in almost all tasks. This underscores the importance of the regularization term in enhancing the alignment between the representation conditions and generated trajectories.

Table 3: Ablation results on the MI regularization term.

| Tasks | W/O MI regularization ($\zeta = 0$) | With MI regularization ($\zeta \neq 0$) |
|---|---|---|
| MT-10 | $67.5 \pm 1.7$ | $\mathbf{68.9 \pm 2.7}$ |
| walker2d-medium-expert | $79.9 \pm 22.6$ | $\mathbf{104.8 \pm 3.0}$ |
| walker2d-medium-replay | $56.5 \pm 6.7$ | $\mathbf{60.5 \pm 1.1}$ |
| hopper-medium-expert | $93.8 \pm 6.8$ | $\mathbf{108.9 \pm 1.0}$ |
| hopper-medium-replay | $54.4 \pm 0.1$ | $\mathbf{56.2 \pm 0.7}$ |
| halfcheetah-medium-expert | $94.4 \pm 1.4$ | $\mathbf{95.0 \pm 2.1}$ |

Regarding that the performance improvement is not significant on some tasks, we would like to supplement several analysis. The diffusion model first models the trajectory distribution represented by offline data and then ensures the generation of relatively better trajectories by controlling conditions.

- For the MT-10 task, simple behavior cloning performs reasonably well (as shown in Figure 4), indicating that the basic trajectory distribution is relatively good. Therefore, controlling conditions may not significantly affect the final performance. Similarly, in some simple tasks like halfcheetah-medium-expert or walker2d-medium-replay and hopper-medium-replay, the trajectories learned directly by the diffusion model exhibit weak dependence on conditions. In these cases, MI regularization provides limited improvement.

- However, on tasks with more complex data modes, such as walker2d-medium-expert and hopper-medium-expert, the conditional distribution learned by the diffusion model is more complex, leading to a more pronounced effect of controlling conditions. In this case, we also observe a significant improvement in conditional generation with the addition of MI regularization.

## C.3 Ablation study on the influence of the number of tasks

We have also conducted an ablation study on the number of tasks $K$ and compared the performance on five tasks. The results including average return and average success rate across tasks are shown below.

From these results, it can be observed that as the number of tasks $K$ increases from 3 to 5 to 10, both the average return and success rate in experiments gradually increase. This suggests that when $K$

Table 4: Ablations on the number of tasks.

| Tasks | $K$=3 | $K$=5 | $K$=10 |
|---|---|---|---|
| Reach-v2 | 172.9 | 1209.4 | 3625.7 |
| Pick-Place-v2 | 2.8 | 2.7 | 564.8 |
| Window-Open-v2 | 130.2 | 1195.6 | 3188.7 |
| Push-v2 | - | 32.2 | 1083.3 |
| Drawer-Open-v2 | - | 586.2 | 2304.0 |
| Average Success Rate Across 5 Tasks | 0.0% | 4.0% | 51.2% |

is not large, increasing the number of tasks $K$ can enhance the performance of a specific task. We attribute this to the speculation that as the number of tasks increases, the representation learning for a specific task improves. As mentioned in Section 3.1, our method learns a task-specific preference representation and an optimal representation by constructing positive and negative samples, where the negative samples include all trajectories from other tasks. When $K$ increases, the number of negative samples significantly increases, which helps our method learn the preference representation under the given task. Therefore, learning from more tasks leads to better representations of a specific task, resulting in improved performance and accuracy for that particular task.

It is also important to note that when $K$ becomes too large, the learning process for the diffusion model becomes more challenging. This is because the diffusion model needs to fit the trajectory distributions of multiple tasks. Especially when the multi-task trajectory data are low-quality, the difficulty of fitting for the diffusion model increases.

## D   Supplementary Results of The Average Performance in MT-10 Tasks

In this section, we present the complete results of our evaluations. All results are obtained over multiple random seeds. All methods are trained with the same data and adopt the same evaluation metric. Tables 5 and 6 elaborate on the success rates on MT-10 tasks given near-optimal and sub-optimal datasets. A high success rate indicates the model's proficiency in consistently accomplishing the tasks in MT-10 benchmarks.

Table 5: Average success rates given near-optimal datasets.

| Environments | MTBC | MTIQL | MTDiff | MT-OPPO-p | MT-OPPO-w | CAMP |
|---|---|---|---|---|---|---|
| button-press-topdown-v2 | $69.0 \pm 1.0$ | $78.0 \pm 0.0$ | $66.7 \pm 20.5$ | $0.0 \pm 0.0$ | $0.0 \pm 0.0$ | $76.0 \pm 12.0$ |
| door-open-v2 | $99.0 \pm 1.0$ | $98.0 \pm 0.0$ | $93.3 \pm 9.4$ | $0.0 \pm 0.0$ | $0.0 \pm 0.0$ | $92.0 \pm 4.0$ |
| drawer-close-v2 | $100.0 \pm 0.0$ | $100.0 \pm 0.0$ | $100.0 \pm 0.0$ | $92.0 \pm 7.5$ | $98.0 \pm 4.0$ | $100.0 \pm 0.0$ |
| drawer-open-v2 | $68.0 \pm 6.0$ | $20.0 \pm 4.0$ | $83.3 \pm 4.7$ | $0.0 \pm 0.0$ | $0.0 \pm 0.0$ | $27.0 \pm 19.0$ |
| peg-insert-side-v2 | $54.0 \pm 12.0$ | $77.0 \pm 1.0$ | $73.3 \pm 12.5$ | $0.0 \pm 0.0$ | $0.0 \pm 0.0$ | $89.0 \pm 1.0$ |
| pick-place-v2 | $6.0 \pm 2.0$ | $3.0 \pm 1.0$ | $76.7 \pm 12.5$ | $0.0 \pm 0.0$ | $0.0 \pm 0.0$ | $70.0 \pm 26.0$ |
| push-v2 | $9.0 \pm 1.0$ | $25.0 \pm 1.0$ | $20.0 \pm 14.1$ | $16.0 \pm 15.0$ | $0.0 \pm 0.0$ | $21.0 \pm 7.0$ |
| reach-v2 | $67.0 \pm 7.0$ | $86.0 \pm 4.0$ | $70.0 \pm 21.6$ | $16.0 \pm 10.2$ | $8.0 \pm 7.6$ | $32.0 \pm 4.0$ |
| window-close-v2 | $100.0 \pm 0.0$ | $94.0 \pm 2.0$ | $100.0 \pm 0.0$ | $0.0 \pm 0.0$ | $30.0 \pm 36.9$ | $100.0 \pm 0.0$ |
| window-open-v2 | $50.0 \pm 32.0$ | $76.0 \pm 2.0$ | $83.3 \pm 4.7$ | $22.0 \pm 13.3$ | $18.0 \pm 18.3$ | $32.0 \pm 4.0$ |
| Average | $62.2 \pm 6.2$ | $65.7 \pm 1.5$ | $76.7 \pm 3.9$ | $14.6 \pm 4.6$ | $15.4 \pm 6.7$ | $68.9 \pm 2.7$ |

Table 6: Average success rates given sub-optimal datasets.

| Environments | MTBC | MTIQL | MTDiff | MT-OPPO-p | MT-OPPO-w | CAMP |
|---|---|---|---|---|---|---|
| button-press-topdown-v2 | $50.0 \pm 6.0$ | $70.0 \pm 2.0$ | $53.3 \pm 9.4$ | $0.0 \pm 0.0$ | $0.0 \pm 0.0$ | $87.2 \pm 12.9$ |
| door-open-v2 | $47.0 \pm 19.0$ | $75.0 \pm 1.0$ | $26.7 \pm 12.5$ | $0.0 \pm 0.0$ | $0.0 \pm 0.0$ | $71.2 \pm 8.8$ |
| drawer-close-v2 | $100.0 \pm 0.0$ | $99.0 \pm 1.0$ | $100.0 \pm 0.0$ | $100.0 \pm 0.0$ | $88.0 \pm 24.0$ | $100.0 \pm 0.0$ |
| drawer-open-v2 | $22.0 \pm 22.0$ | $4.0 \pm 0.0$ | $40.0 \pm 16.3$ | $0.0 \pm 0.0$ | $0.0 \pm 0.0$ | $6.0 \pm 4.9$ |
| peg-insert-side-v2 | $2.0 \pm 2.0$ | $59.0 \pm 5.0$ | $23.3 \pm 4.7$ | $0.0 \pm 0.0$ | $0.0 \pm 0.0$ | $61.6 \pm 26.1$ |
| pick-place-v2 | $1.0 \pm 1.0$ | $2.0 \pm 2.0$ | $0.0 \pm 0.0$ | $4.0 \pm 8.0$ | $0.0 \pm 0.0$ | $0.8 \pm 1.6$ |
| push-v2 | $3.0 \pm 3.0$ | $13.0 \pm 7.0$ | $23.3 \pm 18.9$ | $12.0 \pm 9.7$ | $0.0 \pm 0.0$ | $10.8 \pm 6.5$ |
| reach-v2 | $24.0 \pm 4.0$ | $79.0 \pm 1.0$ | $40.0 \pm 8.2$ | $10.0 \pm 10.9$ | $2.0 \pm 4.0$ | $44.8 \pm 19.1$ |
| window-close-v2 | $47.0 \pm 27.0$ | $76.0 \pm 2.0$ | $63.3 \pm 12.5$ | $0.0 \pm 0.0$ | $2.0 \pm 4.0$ | $98.0 \pm 4.0$ |
| window-open-v2 | $24.0 \pm 2.0$ | $90.0 \pm 0.0$ | $70.0 \pm 14.1$ | $18.0 \pm 21.4$ | $14.0 \pm 18.5$ | $81.6 \pm 12.1$ |
| Average | $32.0 \pm 8.6$ | $56.7 \pm 2.1$ | $44.0 \pm 2.2$ | $14.4 \pm 5.0$ | $10.6 \pm 5.0$ | $56.2 \pm 9.6$ |

# E   Additional Explanations About The Problem Setting

Our method lies in the broad field of Offline Preference Learning, where the agent learns policies from preference data rather than a designed reward function. However, the preference labels defined by scripted teachers or humans are often tailored for a specific task, and learned policies can only align with this task. To solve this problem, we aim to provide a unified preference representation for both **Single- and Multi-Task Preference Learning**. Based on the representation, we learn **Multi-Task Diffusion Policy** via conditional trajectory generation by using the learned representation as a condition.

# F   Acceleration Sampling for Diffusion models

While multi-step denoising process in the diffusion model generation is time-consuming, our contribution is orthogonal to those sampling acceleration methods for diffusion models, such as DDIM [57], DPM-solver [58], EDP [59], and can be easily combined with them. In fact, we have incorporated the implementation of the DPM-solver into our method.. This improvement has boosted the inference speed by a factor of 8.4 compared to the previous method. Specifically, on an Nvidia RTX3090, the denoising time per step has been reduced from 15.2ms to 1.8ms. However, our initial results indicate a compromise in the quality of the generated trajectories, likely due to the complexity of generating continuous trajectories compared to image-generation tasks. We will explore further enhancements to balance performance and quality in our future work.

# G   Further Analysis About MI Regularization

**The conditional diffusion model is not enough to provide alignment**   The remarkable success of Stable Diffusion and Midjourney has highlighted the potential of diffusion models to generate desired images given textual conditions. However, when diffusion models are applied to offline reinforcement learning and trajectory modeling, the situation appears different. Previous work on return-conditioned diffusion models [5, 14] generates trajectories by conditioning on target returns, but their conditions are often hyperparameters adjusted for each environment. As shown in Figure 1, our tests with different conditions reveal that the relationship between the generation and the conditions is not as expected. Given a higher target return, the diffusion model cannot generate better trajectories. In fact, this phenomenon has been noted in other related works as well [60]. Therefore, the current conditional diffusion models are insufficient to provide the necessary alignment between preference and trajectory in our setting. This is one of the main problems we aim to solve.

**Why using MI regularization?**   Inspired by the work in the field of image generation, such as InfoVAE [28], InfoGAN [29], and InfoDiffusion [21], we propose the adoption of mutual information regularization. During the learning phase of the diffusion model, it effectively estimates prior noise, with its posterior estimation conditioned on preference representation. In our debugging phase, we observed that noise estimates from previous works often disregard conditioning, leading to the diffusion model's inability to effectively learn the conditional distribution $p(\tau|c)$. Therefore, in our work, we imposed a constraint of maximizing mutual information on the diffusion model, ensuring a tighter relationship between the posterior noise estimation and the condition. This guarantees that the noise estimation network can capture information about the condition. Consequently, during the denoising generation phase, the trajectories generated by the diffusion model can well correspond to the condition. Moreover, in response to the inefficiency of the multi-step denoising process in diffusion models, we have introduced a reasonable approximation that simplifies the implementation of regularization. The experimental results also demonstrate that enhancing this connection leads to better outcomes than not doing so.

# H   Additional Analysis About The Generalization Ability

## H.1   How to obtain $w_k^*$ for a new task $k$?

Since our method projects trajectory segments into representation space, obtaining $w_k^*$ of the new task $k$ is akin to locating its position in the representation space. Assuming the optimal representations for

known tasks $i$ and $j$ as $w_i^*$ and $w_j^*$ respectively, we can approximate $w_k^*$ with $w_i^*$ or use $w_i^*$ as a starting point to estimate $w_k^*$ when task $k$ is very similar to task $i$, indicating that $w_k^*$ is close to $w_i^*$ in the representation space. Similarly, when task $k$ is composed of task $i$ and task $j$, interpolation between $w_i^*$ and $w_j^*$ suffices to estimate $w_k^*$. These methods are relatively straightforward and direct. However, if task $k$ is distant from known tasks, we consider relearning $w_k^*$ from scratch using trajectory data of task $k$.

Learning the optimal representations $w_k^*$ for new tasks from scratch is consistent with learning representations for known tasks. We require trajectory data with preference pairs on the new task. By using the multi-task preference proposed in Section 3.1, we construct favorable trajectories as positive samples and unfavorable trajectories along with trajectories from other tasks as negative samples. We learn preference representations and optimal representations for task $k$ in the same manner of learning $w_i^*$. Several points should be noted:

- When continuing training on the already trained representation network $f_\psi$, the number of learning samples for the new task is significantly fewer, approximately only one-fourth.

- Learning $w_k^*$ for new tasks can be decoupled from learning the diffusion model. During inference on new tasks, the trained $w_k^*$ can be inputted into the diffusion model to generate target trajectories for the new task.

- Learning $w_k^*$ does not require expert trajectories of task $k$, and mixed data is often sufficient.

## H.2 Dataset requirement?

Our learned diffusion model is designed to model the relationship between preference representations and trajectory segments, rather than directly mapping representations to complete trajectories. The preference representations we learn effectively map different trajectory segments into a representation space and then generate trajectory segments based on the conditions of the representation. Therefore, as long as there are some favorable segments within near-optimal trajectories, the learned distribution will involve optimal trajectories, and we can condition the diffusion model on $w^*$ to generate optimal trajectory segments. In other words, our approach does not require expert trajectories. In fact, the experimental results in Figure 4 validate that our method performs well on both near-optimal and sub-optimal data. Nevertheless, If the quality of the offline datasets is extremely poor, for example, if they consist entirely of random data, it becomes challenging to derive informative preferences. Consequently, we cannot learn a representation space capable of clearly distinguishing the quality of trajectory segments, thereby limiting the generalization ability of the diffusion model. We note that poor performance and generalization on random data is also commonly encountered in offline RL [4] and offline preference learning methods [15, 43].

## H.3 What enables the generalization to unseen tasks?

We attribute this generalization ability to two main factors: the representation space constructed from preference representations learned from multi-task preferences, and the mutual information regularization applied to the conditional diffusion model.

Firstly, by constructing multi-task preferences, we ensure that trajectory segments from different tasks map to distinct regions in the representation space, and segments of different qualities from the same task distribute smoothly in the representation space. This enables our representation space to effectively differentiate between trajectories of different qualities across tasks. Therefore, for a new task $k$, if it is highly similar to known tasks, we can approximate or interpolate $w_k^*$ from the optimal representations of known tasks. Conversely, if it is significantly different, we can reconstruct preference pairs and samples to map the trajectory segments of the new task to different regions in the representation space. Similarly, task $k$ can also be distinguished from other tasks.

Second, we have enhanced the existing conditional diffusion model by introducing MI regularization, strengthening the connection between generated trajectories and given conditions. MI regularization sharpens the correspondence between the representation space and the trajectory segment distribution. In other words, we can control which task the generated trajectories come from by switching the given conditions. When $w_k^*$ for the new task $k$ is obtained through various methods, including approximation, interpolation, or relearning, we can ensure that the diffusion model conditioned on $w_k^*$ generates the desired trajectories for task $k$.

# I Space and Time Complexity

The computation in our method primarily involves two parts: 1) Training the MI Regularized Diffusion Model: Estimating the mutual information between the generated trajectory $\tau_0$ and the preference representation $w$ is computationally intensive, as generating $\tau_0$ requires $k$-step denoising. To mitigate this, we apply approximations and replace $I(\tau_0; w)$ with $I(\tau_k; w)$, thus bypassing the denosing process during training. $I(\tau_k; w)$ can be obtained from the estimated noise at each step, requiring only a two-layer MLP to align the dimensions with $w$. 2) Inference Stage: Sampling during the inference stage is time-consuming, which we have discussed in Appendix F. To address this, we have experimented with DPM-solver [1] to reduce the sampling time of the diffusion model. This improvement has boosted the inference speed by a factor of 8.4 compared to the previous method. Specifically, on an Nvidia RTX3090, the denoising time per step has been reduced from 15.2ms to 1.8ms. Despite the improvement in efficiency, our initial results indicate a compromise in the quality of the generated trajectories, likely due to the complexity of generating continuous trajectories. We will explore further enhancements to balance efficiency and quality in our future work.

We have also conducted a further comparison of the space complexity and time complexity of CAMP, with the diffusion model-based multitask learning method, MTDiff. According to the results in Table 7, the runtime for each update in MTDiff is approximately half of that in CAMP. This is because, in addition to optimizing the diffusion model, CAMP also optimizes the preference representation network and the optimal preference representation. It is worth noting that the learning of preference representation and the optimization of the diffusion model can be decoupled; if decoupled, our method might exhibit greater flexibility and efficiency. On the other hand, by comparing the GPU memory usage during algorithm execution, we found that the space required by CAMP is about one-third of that required by MTDiff. This indicates that, although CAMP is more computationally intensive, its memory space requirements are significantly lower. This reason may be that CAMP utilizes a U-Net based noise predictor, while MTDiff needs a more complex Transformer-based noise predictor.

Table 7: Comparison of MTDiff and CAMP

|                      | MTDiff | CAMP |
|----------------------|--------|------|
| Runtime per update (s) | 32.0   | 62.5 |
| Memory used (MB)     | 7322   | 2516 |

# J Broader Impacts

This paper presents a novel approach to sequential decision-making, aiming to advance the field of machine learning. Our method addresses critical challenges in aligning decision-making processes with human intents and achieving versatility across various tasks. By adopting multi-task preferences as a unified condition for decision-making, we offer a framework that enhances both alignments with human preferences and versatility across tasks.

The societal consequences of our work are manifold. By improving the controllability of decision-making processes concerning human preferences, our approach holds promise for applications in diverse domains such as robotics, healthcare, and personalized recommendation systems. Additionally, by addressing limitations in existing methods regarding the formulation of reward functions, we pave the way for more robust and adaptable machine learning systems.

Further, we believe our work is beneficial for the broader research community in preference learning. For instance, in the alignment of large language models, researchers are increasingly recognizing the importance of diverse human preferences for multi-objective preferences[61–65]. Our work actually offers a perspective on this issue. For text-to-image or text-to-video generation, better alignment with human preferences is also a critical consideration[66], and our work holds relevant implications for these fields.

