# OpenReview forum: "Regularized Conditional Diffusion Model for Multi-Task Preference Alignment"
_NeurIPS.cc/2024/Conference — NeurIPS 2024 poster_

### Official Review · Reviewer_Y24x · 2024-06-28

**Soundness:** 2
**Presentation:** 3
**Contribution:** 3
**Rating:** 5
**Confidence:** 2

**Summary:**

This paper presents a novel approach using a regularized conditional diffusion model to align with preferences in multi-task reinforcement learning (RL) scenarios. The proposed method generates preference representations that effectively distinguish between various task trajectories and returns. By maximizing mutual information between conditions and generated trajectories, the model regularizes existing diffusion frameworks, enabling it to produce optimal task-specific trajectories aligned with given preferences.

**Strengths:**

- The motivation is clear - generating preference representations for various tasks to differentiate them and taking them as the conditions in the diffusion model so that we could consider preference learning in the multi-task scenario.
- The method and all the designs in this method are reasonable to me.

**Weaknesses:**

My main concern is about the results. It seems that CAMP doesn't perform better than existing methods. For details, please refer to the Questions part. Thanks.

**Questions:**

- Could you provide an additional ablation study on the influence of the number of tasks? Specifically, does the representation for task A differ when learning tasks {A, B} together versus tasks {A, B, C} together? Additionally, does the number of tasks impact the accuracy of task A?
- Could you highlight the best results in each row of Table 1? It appears that CAMP's performance is not consistently better than existing methods across these environments, and its average result is not particularly strong. Does this imply that CAMP struggles to learn a good task representation? Alternatively, do you have any other explanations for these observations?
- Regarding the multi-task dataset case, Tables 4 and 5 indicate that CAMP performs well on some tasks but poorly on others. Could you explain why this is the case? Is it due to an imbalance in the training of the multi-task scenario?

---

> ### Author Rebuttal · Authors · 2024-08-06
>
> Thanks for taking the time to provide feedback on our paper. We appreciate your valuable comments and would like to address each of your concerns.
>
> **1. Could you provide an additional ablation study on the influence of the number of tasks? Specifically, does the representation for task A differ when learning tasks {A, B} together versus tasks {A, B, C} together? Additionally, does the number of tasks impact the accuracy of task A?**
>
> Thanks for your insightful suggestion. We have conducted an ablation study on the number of tasks $K$ and compared the performance on five tasks. The results including average return and average success rate across tasks are shown below.
>
> ||Tasks| K=3| K=5| K=10|
> |--|--|--|--|--|
> ||Reach-v2|172.9 |1209.4 |3625.7 |
> ||Pick-Place-v2|2.8|2.7|564.8|
> |Average Return|Window-Open-v2|130.2|1195.6|3188.7|
> ||Push-v2|-|32.2|1083.3|
> ||Drawer-Open-v2|-|586.2|2304.0|
> |Average Success Rate across 5 tasks|-|0.0% | 4.0%| 51.2%|
> ||
>
> From these results, it can be observed that as the number of tasks $K$ increases from 3 to 5 to 10, both the average return and success rate in experiments gradually increase. This suggests that when $K$ is not large, increasing the number of tasks $K$ can enhance the performance of a specific task. We attribute this to the speculation that as the number of tasks increases, the representation learning for a specific task improves. As mentioned in Section 3.1, our method learns a task-specific preference representation and an optimal representation by constructing positive and negative samples, where the negative samples include all trajectories from other tasks. When $K$ increases, the number of negative samples significantly increases, which helps our method learn the preference representation under the given task. Therefore, learning from more tasks leads to better representations of a specific task, resulting in improved performance and accuracy for that particular task.
>
> It is also important to note that when $K$ becomes too large, the learning process for the diffusion model becomes more challenging. This is because the diffusion model needs to fit the trajectory distributions of multiple tasks. Especially when the multi-task trajectory data are low-quality, the difficulty of fitting for the diffusion model increases.
>
>
> **2. Could you highlight the best results in each row of Table 1? It appears that CAMP's performance is not consistently better than existing methods across these environments, and its average result is not particularly strong. Does this imply that CAMP struggles to learn a good task representation? Alternatively, do you have any other explanations for these observations?**
>
>
> Thanks for your feedback. We will highlight the best results in each row of Table 1 to clarify the comparisons in the next version. We notice that your concern is similar to the second question raised by Reviewer 3Xmu. Therefore, **we have provided a detailed and unified response in the Global Rebuttal**. Here, we would like to briefly clarify that in Table 1, the reason CAMP does not appear to outperform all other methods is not due to CAMP's failure to learn a good representation. Rather, it is because we included the reward-based method IQL as a baseline. In the single-task setting, CAMP learns solely from preference data, while IQL learns from actual rewards, making it more challenging for CAMP to learn an optimal policy compared to IQL. However, when comparing CAMP with other preference-based methods like PT and OPPO, we find that CAMP still demonstrates significant advantages.
>
> **3. Regarding the multi-task dataset case, Tables 4 and 5 indicate that CAMP performs well on some tasks but poorly on others. Could you explain why this is the case? Is it due to an imbalance in the training of the multi-task scenario?**
>
>
> Thank you for your question. Tables 4 and 5 show the performance of different methods on the MT-10 tasks with near-optimal and sub-optimal data. The inconsistency in CAMP's performance across the 10 tasks—where it performs well on some tasks but poorly on others—is not due to an imbalance in multi-task training. Instead, it reflects the inherent difficulty of the tasks. Some tasks within the MT-10 set are more challenging, while others are relatively easier. This variability in task difficulty explains the differences in performance observed. Additionally, many tasks in MetaWorld have the characteristic that even if the cumulative reward reaches 90% of the maximum reward, the success rate remains 0. This also contributes to the large variance observed in the results presented in Tables 4 and 5.
>
> **We hope that our responses above, as well as those in the global rebuttal, have addressed your concerns. If they have, we kindly ask you to consider improving our score. If not, we welcome further discussion.**

---

> > ### Comment · Reviewer_Y24x · 2024-08-09
> > **Response to the author**
> >
> > Thanks for the response.
> > Q1: Can I understand that the non-significant superiority of CAMP in Table is because the baseline IQL is too strong? If so, why do we just use IQL?
> > Q2: Do we have any solution for the different difficulties of the tasks in MTL? It's indeed the imbalance problem in MTL and should be solved carefully.

---

> > > ### Author Response · Authors · 2024-08-10
> > >
> > > Thanks for your valuable feedback.
> > >
> > > **A1**: We do not intend to suggest that IQL is excessively strong. Instead, it serves as a representative reward-based offline RL method that can be extended to multi-task scenarios.  The performance discrepancy between IQL and CAMP primarily arises from their different utilization of reward signals and preference data. We compare CAMP with IQL to demonstrate that our approach, which relies solely on preference data, can achieve performance comparable to reward-based methods in multi-task scenarios.
> > >
> > > **A2**: Thanks for your correction. The varying difficulty levels among tasks indeed lead to training imbalances in MTL. In our current work on using diffusion models for multi-task trajectory generation, we did not account for conflicts between tasks, which could result in uneven performance in MetaWorld. To address this problem, previous MTL methods have employed strategies such as weighted summation of multi-task losses, gradient manipulation of each loss , and Pareto optimal solutions. These strategies optimize multiple losses simultaneously and may prevent performance degradation due to task conflicts. However, these approaches mainly focus on vision tasks and have been barely investigated in diffusion models or trajectory generation for decision-making. Our method can be combined with these strategies to further enhance multi-task decision-making performance. Recently, several methods, including uncertainty weighting [1] and PCgrad [2], have been integrated to mitigate the negative transfer problem in image generation using diffusion models [3]. These approaches can provide valuable guidance for future work and be extended to our mutli-task decision-making scenarios, mitigating the imbalance in our conditional trajectory generation.
> > >
> > > [1] Kendall A, Gal Y, Cipolla R. Multi-task learning using uncertainty to weigh losses for scene geometry and semantics, CVPR 2018.
> > >
> > > [2] Yu T, Kumar S, Gupta A, et al. Gradient surgery for multi-task learning, NeurIPS 2020.
> > >
> > > [3] Go H, Lee Y, Lee S, et al. Addressing negative transfer in diffusion models, NeurIPS 2023.

---

> ### Author Response · Authors · 2024-08-13
> **Looking forward to your feedback**
>
> Dear Reviewer Y24x,
>
> We first would like to thank the reviewer's efforts and time in reviewing our work. We were wondering if our responses have resolved your concerns. Since the discussion period is ending soon, we would like to express our sincere gratitude if you could check our reply to your comments. We will be happy to have further discussions with you if there are still some remaining questions! We sincerely look forward to your kind reply!
>
> Best regards,
>
> The authors

---

### Official Review · Reviewer_3Xmu · 2024-07-10

**Soundness:** 3
**Presentation:** 3
**Contribution:** 3
**Rating:** 6
**Confidence:** 3

**Summary:**

The paper proposes a novel method called the Regularized Conditional Diffusion Model for multi-task preference alignment, addressing the challenge of generating trajectories that align with human preferences in multi-task settings. This method introduces preference-based representations and a mutual information regularization technique to improve the alignment between generated trajectories and preferences.

**Strengths:**

1.	The method effectively captures and utilizes versatile preference representations, allowing for better alignment with human intents across multiple tasks.
2.	The introduction of a mutual information regularization objective enhances the consistency between conditions and generated trajectories, improving performance.
3.	The diffusion model guided by learned preference representations generates trajectories that align closely with desired preferences.

**Weaknesses:**

1.	The authors’ proposed method shares a similar motivation with OPPO. In the appendix, they explain the differences between their method and other approaches like OPPO and MTDiff. Their method improves upon OPPO and MTDiff by incorporating Diffusion and other techniques such as regularization for multi-task preferences. It is recommended that the authors emphasize these distinctions in the main text and update the contribution list to highlight their unique contributions.
2.	From the comparative experimental results presented in the figures and tables, it is evident that the authors’ method does not achieve the best performance across all metrics. It is suggested that the authors provide a corresponding discussion instead of merely describing the contents of the figures and tables.
3.	The authors’ method involves computationally intensive modules such as mutual information and diffusion models. It is advised to include a discussion on the computational costs of the model, such as space complexity and time complexity, to enable a comprehensive comparison with related models.
4.	Given that the authors’ model addresses multi-task preferences, it would be beneficial to discuss the impact of the quality of the provided preference data on the results.

**Questions:**

Please see the weaknesses

---

> ### Author Rebuttal · Authors · 2024-08-06
>
> Thanks for taking the time to provide feedback on our paper. We appreciate your valuable comments and would like to address each of your concerns.
>
> **1. It is recommended that the authors emphasize these distinctions and update the contribution list.**
>
>
> Thanks for your suggestion. We will emphasize the distinctions between CAMP, OPPO, and MTDiff in the main text and update the contribution list in the next version. Specifically, our contributions can be summarized as follows:
>
> - We have introduced multi-task preferences and proposed to learn high-dimensional preference representations from trajectory segments, which extends the versatility of traditional preference-based methods like OPPO.
> - We have proposed a mutual information regularized diffusion model for multi-task trajectory generation by conditioning on preference representations, saving the need for actual rewards and expert prompts used in MTDiff.
> - We have conducted extensive experiments on MetaWorld, showing comparable performance to reward-based methods, superiority to baseline preference-based methods, and favorable generalization performance to unseen tasks.
>
>
> **2. The proposed method does not achieve the best performance across all metrics. It is suggested that the authors provide a corresponding discussion.**
>
> Thanks for your question. We found that this issue is quite similar to the second question raised by Reviewer Y24x. Therefore, **we have provided a detailed and unified response in the Global Rebuttal**. Please refer to it for more information. Here, we can briefly summarize some conclusions: preference data is not as accurate and abundant as reward data. Hence, methods that learn from preference data are less likely to perform as well as reward-based methods. Our experiments in both multi-task and single-task settings aim to demonstrate that CAMP, using only preference data, can achieve performance comparable to the best reward-based methods, while significantly outperforming other preference-based methods.
>
> **3. It is advised to include a discussion on the computational costs of the model, such as space complexity and time complexity.**
>
> Thanks for your insightful question. Regarding the concerns of the reviewer about the optimization of mutual information and diffusion models, we have proposed corresponding improvement methods. These include scaling and simplifying the original mutual information objective, as well as employing faster sampling methods for diffusion models. **Please refer to our response to Question 2 from Reviewer L2wb for more details**, where we have discussed which parts of our approach were more time-consuming and how CAMP was optimized to address these issues.
>
> Here, we have conducted a further comparison of the space complexity and time complexity of CAMP, with the diffusion model-based multitask learning method, MTDiff [1]. Both methods were run on an Nvidia RTX 4090 GPU.
>
> | | MTDiff | CAMP |
> |--| --|--|
> |Runtime per update (s)| 32.0| 62.5|
> |Memory used (MB)|7322|2516|
> |
>
> According to the above results, the runtime for each update in MTDiff is approximately half of that in CAMP. This is because, in addition to optimizing the diffusion model, CAMP also optimizes the preference representation network and the optimal preference representation. It is worth noting that the learning of preference representation and the optimization of the diffusion model can be decoupled; if decoupled, our method might exhibit greater flexibility and efficiency. On the other hand, by comparing the GPU memory usage during algorithm execution, we found that the space required by CAMP is about one-third of that required by MTDiff. This indicates that, although CAMP is more computationally intensive, its memory space requirements are significantly lower. This reason may be that CAMP utilizes a U-Net based noise predictor, while MTDiff needs a more complex Transformer-based noise predictor.
>
> [1] He H, Bai C, Xu K, et al. Diffusion model is an effective planner and data synthesizer for multi-task reinforcement learning. NeurIPS 2023
>
> **4. It would be beneficial to discuss the impact of the quality of the provided preference data on the results.**
>
> Thanks for your suggestion. We would like to discuss the impact of the quality of preference data from two perspectives: noisy preferences and the quality of provided trajectory data. Firstly, noisy preferences can indeed influence the accuracy of learned preference representations. To make representation learning robust to noisy preferences, our method has incorporated two useful components: 1) inspired by contrastive learning methods [1], our method also constructs positive and negative samples and optimizes the representation network so that they can be separated in the representation space. This is more robust than learning a reward model aligned with preferences in traditional preference-based RL  methods. 2) Our method learns a representation distribution, which captures the uncertainty contained in the preference dataset [2].
>
> Secondly, in multi-task learning experiments, we tested learning results using near-optimal and sub-optimal datasets, which vary in the quality of trajectory data. The near-optimal datasets, which contain more expert trajectories, have higher quality compared to sub-optimal datasets. Figure 4 illustrates that our method is robust to different qualities of trajectory data.
>
> [1] Shen W, Zhang X, Yao Y, et al. Improving Reinforcement Learning from Human Feedback Using Contrastive Rewards. ArXiv:2403.07708.
>
> [2] Duan S, Yi X, Zhang P, et al. Negating negatives: Alignment without human positive samples via distributional dispreference optimization. ArXiv:2403.03419.
>
> **We hope that our responses above, as well as those in the global rebuttal, have addressed your concerns. If they have, we kindly ask you to consider improving our score. If not, we welcome further discussion.**

---

> ### Author Response · Authors · 2024-08-13
> **Looking forward to your feedback**
>
> Dear Reviewer 3Xmu,
>
> We first would like to thank the reviewer's efforts and time in reviewing our work. We were wondering if our responses have resolved your concerns. Since the discussion period is ending soon, we would like to express our sincere gratitude if you could check our reply to your comments. We will be happy to have further discussions with you if there are still some remaining questions! We sincerely look forward to your kind reply!
>
> Best regards,
>
> The authors

---

> > ### Comment · Reviewer_3Xmu · 2024-08-13
> >
> > Thanks for your responses and clarifications. It looks good to me. May I ask a further question? Could you please provide more details about constructing positive and negative samples for this work? Thanks.

---

> > > ### Author Response · Authors · 2024-08-14
> > >
> > > Thanks for your valuable feedback. The construction of positive and negative samples is illustrated in Figure 2. Here, we provide a further explanation.
> > >
> > > In the $m$ tasks, before calculating the loss and updating the gradients each time, a task $i \in [m]$ is randomly sampled. Then, from the trajectory data of task $i$, two batches of data with batch size are sampled. These are divided into two groups based on their preferences, with one group having higher preference and the other having lower preference. The batch with higher preference is taken as the positive samples, and the batch with lower preference as the negative samples. Additionally, trajectory data with batch size are sampled from all other tasks $j \in [m], j \neq i$, and all these trajectories are considered as negative samples in this gradient update. In summary, for each update, $(m+1)\times batchsize$ trajectories are sampled, where the batch size trajectories are positive samples from the sampled task $i$, and $m\times batchsize$ trajectories are negative samples from task $i$ and other tasks $j$.
> > >
> > > We hope that our responses have addressed your concerns.

---

> > > > ### Comment · Reviewer_3Xmu · 2024-08-14
> > > >
> > > > Thank you very much for the further clarifications. I will increase my score.

---

> > > > > ### Author Response · Authors · 2024-08-14
> > > > >
> > > > > We are glad that our response has addressed your concerns, and we commend your effort in evaluating our paper and raising the score. Your valuable suggestions are greatly appreciated and very helpful in improving the quality of our paper. We will incorporate discussions on contributions, experiments, and computational complexity in the next version based on your suggestions.

---

### Official Review · Reviewer_L2wb · 2024-07-12

**Soundness:** 3
**Presentation:** 3
**Contribution:** 3
**Rating:** 6
**Confidence:** 4

**Summary:**

This paper proposes a regularized conditional diffusion model which aligns with preferences for multi-task scenarios. To achieve this, learnable representations for preferences are aligned with preference labels, which can be adopted as condition inputs to guide the generation process of diffusion models. Meanwhile, in order to keep consistency between conditions and generated trajectories, a mutual information regularization method is utilized to improve their alignment with preferences. To verify its effectiveness, extensive experiments have been conducted on two datasets D4RL & Meta-World which cover both single- and multiple-task scenarios, and the results demonstrate the superior performance and generalization ability of the proposed method in aligning preferences across different tasks.

**Strengths:**

1. This paper is well-written and detailed. The authors provide sufficient technical principles and experimental details for readers to understand and be able to reproduce their work.
2. This paper gives a novelty idea in human preferences alignment and combines it with diffusion models in multi-task scenarios, which can be conducive to further foundation models finetuning.
3. Conducting extensive experiments on public datasets and comparing them with different state-of-the-art methods makes the proposed method more practical and convincing.

**Weaknesses:**

1. From Table 1, we can see a large variation in performance for different tasks with the same implementation method.
2. The proposed method seems to have high computational complexity.
3. From Table 2, although the generalization ability for unseen tasks is somewhat improved by this approach compared to the baseline methods, the average performance is still not very good.

**Questions:**

1. I wonder what factors contribute to this phenomenon and how we choose the optimal approach for a given task.
2. I would like to know whether the method proposed in the paper receives a greater impact on efficiency as the type and number of tasks increase.
3. Is the generalization ability of this approach feasible in practical applications?

---

> ### Author Rebuttal · Authors · 2024-08-06
>
> Thanks for taking the time to provide feedback on our paper. We appreciate your valuable comments and would like to address each of your concerns.
>
> **1. Table 1 presents a large variation in performance for different tasks with the same implementation method. What factors contribute to this phenomenon, and how do we choose the optimal approach for a given task?**
>
> Thanks for your questions. **The significant variation in performance for the same method across different tasks, as seen in Table 1, is primarily due to the varying quality of the offline data used during the learning process.** Specifically, the single-task performance is evaluated on the commonly used D4RL Gym Mujoco tasks for offline RL, which include halfcheetah, walker2d, and hopper tasks. We employed three offline datasets: medium, medium-replay, and expert, which are obtained by different behavior policies. The trajectory quality in the medium and medium-replay datasets is lower than that in the expert dataset. Therefore, each method's performance on the medium and medium-replay datasets is generally lower than on the expert dataset.
>
>
> As for how to choose the optimal approach for a given task, it depends on which training data are given. According to Table 1, for single-task learning, CAMP is less effective than IQL which utilizes actual rewards. This is understandable because reward signals provide more detailed and accurate information compared to preference data, and **learning with actual rewards in single tasks can potentially yield better performance**. Despite this, only given preference data, **CAMP outperforms other preference-based methods**. Therefore, if the given task provides actual reward signals, reward-based methods are generally superior. However, if the given task only has preference data, CAMP is better than other preference-based methods.
>
> **2. The proposed method seems to have high computational complexity.**
>
> Thanks for your question. Regarding computation complexity, we have presented specific results for space complexity and time complexity in our response to Question 3 from Reviewer 3Xmu. Please refer to that for details. Here, we analyze the parts of the method that are more time-consuming and introduce how we addressed these issues. The computation in our method primarily involves two parts:
>
> - **Training the MI Regularized Diffusion Model**: Estimating the mutual information (MI) between the generated trajectory $\tau_0$ and the preference representation $w$ is computationally intensive, as generating $\tau_0$ requires $k$-step denoising. To mitigate this, we apply approximations and replace $I(\tau_0;w)$ with $I(\tau_k;w)$, thus bypassing the denosing process during training. $I(\tau_k;w)$ can be obtained from the estimated noise at each step, requiring only a two-layer MLP to align the dimensions with $w$.
> - **Inference Stage**: Sampling during the inference stage is time-consuming, which we have discussed in Appendix F. To address this, we have experimented with DPM-solver [1] to reduce the sampling time of the diffusion model . This improvement has boosted the inference speed by a factor of 8.4 compared to the previous method. Specifically, on an Nvidia RTX3090, the denoising time per step has been reduced from 15.2ms to 1.8ms. Despite the improvement in efficiency, our initial results indicate a compromise in the quality of the generated trajectories, likely due to the complexity of generating continuous trajectories. We will explore further enhancements to balance efficiency and quality in our future work.
>
> [1] Lu C, Zhou Y, Bao F, et al. Dpm-solver: A fast ode solver for diffusion probabilistic model sampling in around 10 steps. NeurIPS 2022
>
> **3. Whether the proposed method receive a greater impact on efficiency as the type and number of tasks increase?**
>
> Thanks for your question. This issue is quite similar to the first question raised by Reviewer Y24x, so **we have included it in the Global Rebuttal. More experimental results can be found in our response to Question 1 from Reviewer Y24x**. Here, we provide a brief conclusion: As the types and number of tasks increase, the learning of preference representations improves, but it becomes more challenging for the diffusion model to fit the trajectory distributions of multiple tasks. The final performance is influenced by both of these factors.
>
> **4. From Table 2, the average performance on the generalization ability for unseen tasks is still not very good.**
>
>
> Thanks for your question. The average performance for unseen tasks is not very good due to the inherent challenge of generalization in MetaWorld, where the difference between the learned and unseen tasks is substantial. This discrepancy causes the preference representations of new tasks learned by CAMP to be distant from those of existing tasks in the representation space. Although we incorporate MI regularization to enable the diffusion model to generate trajectories corresponding to the task preferences and exhibit generalization in the representation space, this generalization ability is weakened when the preference representations of new tasks differ significantly from those of existing tasks. This leads to the performance in Table 2 not being particularly outstanding. Nevertheless, CAMP still demonstrates superior performance compared to existing methods.
>
>
> **5. Is the generalization ability of this approach feasible in practical applications?**
>
> Thanks for your question regarding the feasibility of the generalization ability of this approach in practical applications. While the current results indicate that the generalization ability is not yet at an optimal level, there is potential for improvement. With sufficient high-quality data and a diverse set of tasks, the generalization ability of the model can be further enhanced, making it more feasible for practical applications.

---

> ### Author Response · Authors · 2024-08-13
> **Looking forward to your feedback**
>
> Dear Reviewer L2wb,
>
> We first would like to thank the reviewer's efforts and time in reviewing our work. We were wondering if our responses have resolved your concerns. Since the discussion period is ending soon, we would like to express our sincere gratitude if you could check our reply to your comments. We will be happy to have further discussions with you if there are still some remaining questions! We sincerely look forward to your kind reply!
>
> Best regards,
>
> The authors

---

### Author Rebuttal · Authors · 2024-08-06

# Global Rebuttal

We appreciate all the reviewers for their dedication and recognition of our work, including acknowledging our motivation and novelty [L2wb, Y24x], the effectiveness and improvements of our method [L2wb, 3Xmu], and our presentation [L2wb, Y24x].

We have attempted to address each reviewer's concerns point by point. We notice that the reviewers have some similar questions. In addition to individual responses, we would like to provide unified responses here.

- **Why CAMP's performance is not consistently better than existing methods across all environments?**

Reviewers 3Xmu and Y24x both expressed confusion regarding the phenomenon observed in Figure 4 and Table 1, where our method does not appear to outperform all other methods. We would like to provide further explanations for the experimental results. Figure 4 and Table 1 compare the performance in multi-task and single-task scenarios, respectively. We select **reward-based** methods and **preference-based** methods as our baselines.

In Figure 4, the orange bars represent methods that learn from actual rewards, while the green bars represent methods that learn from preference data. Our aim is to show that in multi-task learning, whether given near-optimal datasets or sub-optimal datasets, our preference-based learning method achieves performance comparable to the best reward-based learning method. Additionally, our method demonstrates a clear advantage over other preference-based learning methods in multi-task scenarios, demonstrating the superiority of learning high-dimensional multi-task preference representations for trajectory generation.

In Table 1, we also distinguish between reward-based methods and preference-based methods. While our method outperforms other preference-based baseline methods, it is slightly weaker than the reward-based method IQL, which utilizes actual rewards for each state-action pair. The inferiority to IQL is because reward signals provide absolute evaluation of state-action pairs as scalars, offering more information compared to relative preferences. CAMP uses relative preferences between pairs of trajectories, which makes learning more challenging.

- **Computational Complexity**

Reviewers L2wb and 3Xmu both raised concerns about the computational complexity of our method. In our response to reviewer L2wb, we analyze two main time-consuming components of CAMP and explain the adjustments we have made to each part, including simplifications to the original MI regularization objective and applications of sampling acceleration methods on the diffusion model. In our response to reviewer 3Xmu, we present additional results and compare CAMP and MTDiff in terms of both space complexity and time complexity. Although CAMP requires a longer update time due to the consideration of representation learning, it needs significantly less GPU memory.

- **What about the performance when the number of tasks increases?**

Reviewers L2wb and Y24x both raised concerns about whether the performance of our method improves as the number of tasks increases, including the accuracy and efficiency of representation learning. This issue can be considered from two perspectives.

On the one hand, when the type and number of tasks increase, the representation learning for each task indeed becomes more efficient. As explained in Section 3.1, our method learns preference representations by constructing positive and negative samples. The positive samples are high-quality trajectories of a specific task, while the negative samples consist of all trajectories from other tasks. As the type and number of tasks increase, the number of negative samples significantly increases, thereby greatly enhancing the efficiency of representation learning. This can be validated by the supplementary results in our response to Question 1 from Reviewer Y24x. When the number of tasks increased from 3 to 5 to 10, the performance of specific tasks indeed improved.

On the other hand, as the number of tasks increases, especially when the trajectory data provided for each task is of low quality, the diffusion model may struggle to fit the trajectory distribution for each task well. In other words, as the number of tasks increases, the learning process for the diffusion model becomes more challenging. This can potentially affect the trajectory generation for each task. The generated trajectories may still perform poorly, even if the preference representations for different tasks are accurate.

---

### Decision · Program_Chairs · 2024-09-25

**Decision:**

Accept (poster)

**Comment:**

Post rebuttal, all reviewers have agreed to accept the paper, with their primary concern regarding non-SOTA performance addressed. After thoroughly reviewing the content and rebuttals, I concur that this paper proposes a valuable regularized conditional diffusion model for multi-task preference alignment. Therefore, I recommend acceptance. Please ensure that the necessary revisions are incorporated into the final version.